



# Landslide initiation thresholds in data sparse regions: Application to landslide early warning criteria in Sitka, Alaska, USA

Annette I. Patton[1,2], Lisa V. Luna [3,4,5], Joshua J. Roering[2], Aaron Jacobs[6], Oliver Korup[3,4], Benjamin B. Mirus[7]

[1]Sitka Sound Science Center, Sitka, Alaska, USA.
[2]University of Oregon, Eugene, Oregon, USA.
[3]Institute of Environmental Science and Geography, University of Potsdam, Potsdam, Germany.
[4]Institute of Geosciences, University of Potsdam, Potsdam, Germany.
[5]Potsdam Institute for Climate Impact Research, Potsdam, Germany.
[6]NOAA National Weather Service Forecast Office Juneau, Alaska, USA.
[7]U.S. Geological Survey, Geologic Hazards Science Center, Golden, Colorado, USA.

*Correspondence to*: Annette Patton (apatton@sitkascience.org) and Lisa Luna (luna@uni-potsdam.de)

**Abstract.** Probabilistic models to inform landslide early warning systems often rely on rainfall totals observed during past events with landslides. However, these models are generally developed for broad regions using large catalogs, with dozens, hundreds, or
even thousands of landslide occurrences. This study evaluates strategies for training landslide forecasting models with a scanty record of landslide-triggering events, which is a typical limitation in remote, sparsely populated regions. We train and evaluate 136 statistical models with a rainfall dataset with five landslide-triggering rainfall events recorded near Sitka, Alaska, USA, as well as >6,000 days of non-triggering rainfall (2002–2020). We use Akaike, Bayesian, and leave-one-out information criteria to compare models trained on cumulative precipitation at timescales ranging from 1 hour to 2 weeks, using both frequentist and
Bayesian methods to estimate the daily probability and intensity of potential landslide occurrence (logistic regression and Poisson regression). We evaluate the best-fit models using leave-one-out validation as well as with testing a subset of the data. Despite this sparse landslide inventory, we find that probabilistic models can effectively distinguish days with landslides from days without. Although frequentist and Bayesian inference produce similar estimates of landslide hazard, they do have different implications for use and interpretation: frequentist models are familiar and easy to implement, but Bayesian models capture the rare-events problem
more explicitly and allow for better understanding of parameter uncertainty given the available data. Three-hour precipitation totals are the best predictor of elevated landslide hazard, and adding antecedent precipitation (days to weeks) did not improve model performance. This relatively short timescale combined with the limited role of antecedent conditions reflects the rapid draining of porous colluvial soils on very steep hillslopes around Sitka. We use the resulting estimates of daily landslide probability to establish two decision boundaries for three levels of warning. With these decision boundaries, the frequentist logistic regression model
incorporates National Weather Service quantitative precipitation forecasts into a real-time landslide early warning "dashboard" system (sitkalandslide.org). This dashboard provides accessible and data-driven situational awareness for community members and emergency managers.

## 1 Introduction

On August 18, 2015, an extreme rain event initiated more than 40 landslides on the islands near Sitka, Alaska, USA, including a
debris flow that resulted in three fatalities (Busch et al., 2021). Over a six-hour period, the Sitka area received 2.5–3.25 inches of rain, and the three-hour storm totals had an estimated 45-year return period. Following this event, the community convened a GeoTask Force to identify priority questions related to landslide risk and hazard (Sitka Sound Science Center, 2016). Community




leaders and technical experts determined the need for a landslide early warning system. This study results from the actions of the community to seek support to reduce landslide risks.


Landslide early warning has the potential to save lives by providing actionable information in advance of an imminent landslide event (e.g., Guzzetti et al., 2020). Landslide early warning systems consist of a prediction ("now-cast" or "forecast") of landslide occurrence, one or more thresholds for action, and a method for disseminating warning information. Decades of investigation around the world have demonstrated the value of using precipitation and hydrologic conditions to predict landslides (e.g., Chae et

al., 2017; Guzzetti et al., 2020), but prediction strategies vary. Most studies determine decision thresholds that aim to separate periods when landslides are likely from periods when they are not. These thresholds may be based on precipitation intensity and duration, consider cumulative precipitation over different time periods (Guzzetti et al., 2008; Bogaard and Greco, 2018; Mirus et al., 2018b), and/or incorporate in situ hydrologic data or estimates of antecedent hillslope hydrological conditions (Mirus et al., 2018a; Thomas et al., 2018; Marino et al., 2020; Wicki et al., 2020). Thresholds may indicate the minimum accumulation of

precipitation needed to initiate landslides or attempt to optimally separate triggering from non-triggering precipitation events (Segoni et al., 2018).

Accurately predicting rare events like landslides is challenging because the complex and spatially heterogenous processes that drive landslide initiation are difficult to characterize at sufficiently high resolution across broad regions. In this study, instead of

trying to predict the *spatial* occurrence, we focus on predicting the *temporal* occurrence of landslides (when and how many failures) within a given study area.

Both empirically and physically based hazard assessments and warning systems require sufficient in situ data to be developed, calibrated, and validated. For example, lack of high-resolution imagery and in situ measurements of parameters such as soil bulk

density, thickness, and hydraulic properties hinders the development of physically based models. Detailed precipitation and hydrologic records with high temporal resolution (hourly or finer) rarely cover long timescales (years to decades). Additionally, remote, sparsely populated areas typically lack inventories of landslide occurrence. These limited datasets of landslide occurrence and associated triggering conditions make it challenging to develop empirical models for landslide initiation, which may have large uncertainties, are often difficult to validate, and cause detrimental false positives in early warning systems. Yet, vulnerability to

landslides is often high in remote and data-sparse regions due to limited infrastructure and access to external aid (Cutter and Finch, 2008). Improving prediction of landslide hazards in remote regions is a critical step to supporting resilient communities.

In this study, we developed a landslide early warning system for the remote community of Sitka, Alaska, USA, (Fig. 1), which had a population of 8,407 in 2021 (U.S. Census Bureau, 2021). We trained statistical models with limited landslide inventory data to

estimate landslide probability and the number of landslides in the study area based on observed and forecasted precipitation, and then used these models to establish thresholds for landslide early warning. We use the term "landslide prediction" to refer to estimates of elevated landslide hazard in the future based on forecasted precipitation.

### 1.2 Study area: Sitka, southeast Alaska

Landslides in southeast Alaska pose persistent hazards to the small, isolated communities that are on the flanks of hillslopes over-

steepened by glaciers. The majority of failures are debris flows initiated by shallow landslides (Swanston and Marion, 1991;





Johnson et al., 2000). Steep hillslopes with thin volcanic soils overlying till are especially susceptible to shallow-seated landslides (Swanston, 1970; Sidle and Swanston, 1981; Patton et al., 2022). Following the fatal debris flow event in Sitka on August 18, 2015 (Busch et al., 2021), community organizers identified the need to better understand both where and when landslides are likely to occur in Sitka.


The landscape surrounding Sitka (Fig. 1) is geomorphically complex (Patton et al., 2022), having been sculpted by tectonic activity (White et al., 2016; Elliott and Freymueller, 2020), Pleistocene glaciation (Hamilton, 1986; Mann, 1986), volcanic eruptions (Riehle et al., 1992b, 1992a), and a long history of human settlement (Sandberg, 2013; Lesnek et al., 2018).

The mid-latitude maritime climate in Sitka is characterized by high annual precipitation. During the last climatic normal (1981–2010), mean annual precipitation at sea level was 2205 mm (Wendler et al., 2016), but steep orographic gradients and complex topography result in spatially heterogenous climate and weather patterns. Mean monthly temperatures stay above freezing all year. Variable snowpacks accumulate in winter months, particularly at high elevations, but most precipitation occurs as non-freezing rain in coastal and low-elevation areas. Rainfall occurs year-round in southeast Alaska, but August-November are the wettest

months.

In particular, atmospheric rivers (ARs) account for 90% of extreme precipitation in southeast Alaska, where "extreme" precipitation was statistically defined using the block maxima approach by identifying one extreme event per year and per season (Sharma and Déry, 2019; Sharma and Dery, 2020). The AR contribution to extreme precipitation is particularly high (>90%) from

September to December. Across southeast Alaska, as well as much of western North America, ARs initiate the vast majority of shallow precipitation-related landslides, although a minority of those ARs actually trigger widespread landsliding (Jacobs et al., 2016; Oakley et al., 2017; Cordeira et al., 2019).

Given this geographic setting, the community of Sitka is exposed to persistent, although largely unquantified, landslide hazards

(Miller, 2019; Busch et al., 2021; Patton et al., 2022). Although it is difficult or impossible to reduce landslide hazards across broad hillslopes, landslide early warning systems can greatly reduce landslide risk to life and safety in these areas. With sufficient warning, residents can voluntarily evacuate high-hazard neighborhoods.



**Figure 1. Study area. (A) Google Earth image of Sitka (Sheet'Ka), Alaska (©Landsat/Copernicus and Maxar Technologies, 2021). The**
**town lies below over-steepened postglacial hillslopes susceptible to landslides. Some of the existing residential and municipal areas are built in landslide initiation or runout zones. (B) Map of recent rain-triggered landslides evaluated in this study shown on a shaded relief map from the U.S. Geological Survey 5-meter digital data, 2014. Higher contrast delineates areas within 2 km of the Sitka road network (dark lines). (C) Photo of the South Kramer Debris Flow, which initiated on August 18, 2015, and resulted in three fatalities. Photograph courtesy of the authors.**






### 1.3 Developing precipitation thresholds for landslide warning

Landslide hazard estimates and precipitation thresholds exist at the global (Guzzetti et al., 2008; Kirschbaum and Stanley, 2018; Khan et al., 2021), regional (Piciullo et al., 2018; Segoni et al., 2018), and local scales (e.g., Mirus et al., 2018a), and even for individual landslides (Kristensen et al., 2021). Developing and applying new thresholds for landslide warning requires determining

the most relevant variables and timescales to model landslide hazard in a particular region, considering data availability, and taking the risk tolerance of the targeted community into account. Almost every investigation of antecedent-triggering precipitation thresholds uses different observation timescales. These differences reflect different landslide types of interest (shallow versus deep-seated), hydrogeomorphic controls, climate, and the type and length of records available.

Hydrometeorological thresholds for landslide initiation have been proposed for nearby remote areas of British Columbia (Jakob et al., 2006) and suburban Vancouver (Jakob et al., 2012) in Canada, but no systematic landslide warning threshold currently exists at either local scales for towns within southeast Alaska or at the regional scale for southeast Alaska as a whole, despite its high susceptibility to slope failures (e.g., Darrow et al., 2022; Patton et al., 2022). Generations of knowledge in southeast Alaska and close observation of the natural environment provide rich understanding of landslides and other natural processes, but southeast

Alaska lacks extensive written records of landslide occurrence with daily timestamps and sub-daily, spatially distributed precipitation records. This contrasts with many well-established landslide prediction models developed in the European Alps, Japan, and other data-rich regions that can draw on tens to thousands of observations of landslide-triggering precipitation and gridded precipitation datasets with high spatial and temporal resolution (e.g., Osanai et al., 2010; Saito et al., 2010; Berti et al., 2012; Lee et al., 2015; Leonarduzzi et al., 2017; Piciullo et al., 2017). Although previous estimates of rainfall thresholds have

included only precipitation events that triggered landslides (Peruccacci et al., 2017), recent research has shown that including records of precipitation that did *not* trigger landslides can help sparse landslide datasets perform well (Peres and Cancelliere, 2021). Warning systems developed from hundreds to thousands of observed landslides are generally considered more trustworthy than those with few landslide-inducing events.

In southeast Alaska, the National Weather Service (NWS) forecasting products provide the best available warning information through weather and hydrologic watches, warnings, and advisories, but both communities and NWS forecasters have expressed a need for systematic analysis of landslide potential under different storm conditions (Busch et al., 2021). Recent investigations in Sitka, Alaska, (Booth et al., 2020; Chu et al., 2021; Vascik et al., 2021) and the community's desire for real-time landslide hazard assessments make this an ideal region to identify new precipitation thresholds and expand on established landslide prediction

techniques for use in data-sparse regions. Our research objective is to provide the community of Sitka with a landslide early warning system that provides real-time and forecasted assessments of landslide hazard to support individual and community-wide decision making. We estimate two metrics of daily landslide hazard in Sitka using statistical models trained with landslide inventory data and precipitation records. As described in detail in the methods section, our approach relies on models developed using hourly precipitation data from both landslide-triggering days (five) and all non-triggering days (>6000) within the period of

record between 2002 and 2020.

### 2 Methods and Data

To develop daily estimates of landslide hazard, we



1. Compiled information about landslide occurrences with known timing near Sitka, Alaska, and weather-station precipitation data for a period of record with hourly precipitation data (2002–2020). Timing of each landslide was known within 12 hours or finer.

2. Trained frequentist and Bayesian models (136 total) with historical records of precipitation and landslides to predict the daily probability of landslides (logistic regression) and the number of landslides (Poisson regression) that could occur based on cumulative precipitation. Logistic regression and Poisson regression are generalized linear models that can incorporate any number of predictor variables, including precipitation at different timescales and groundwater or hydrologic data.

3. Compared the 136 models (Table 1) using a range of cumulative precipitation timescales to select the most appropriate model for the warning system. We considered models with a single predictor (cumulative "triggering" precipitation) and models with two predictors (cumulative "antecedent" precipitation and cumulative "triggering" precipitation during a specific time period on a given day).

4. Checked the most appropriate model's sensitivity (and thus robustness) by removing individual landslide events (leave-one-out/jackknife validation).

5. Using input from Sitka community members, established heuristic decision thresholds for multiple landslide warning levels based on the estimated probability of landsliding and expert judgement.

6. Assessed how well a model trained on an earlier section of the time series was able to predict landslides in a later portion of the time series based on these thresholds, compared its predictive skill to a simpler alternative model based on historical landslide frequency, and evaluated how often landslide warnings would have been issued in the past.

The naming scheme we used for all models (frequentist versus Bayesian, logistic regression versus Poisson regression, precipitation timescales) is summarized in Table 1.

**2.1 Data sources**

To investigate landslide conditions in Sitka, we used existing hourly precipitation records from the nearby weather station (NWS station code PASI) operated by the Federal Aviation Administration (FAA) at Sitka Airport (NOAA NCEI, 2001). Climate records in Sitka go back to the early 19th century (Wendler et al., 2016), and hourly precipitation data (or sub-hourly) have been recorded at the airport weather station since 2002; we included all days with observations between November 12, 2002, and December 13, 2020. A nearby U.S. Climate Reference Network (USCRN) meteorological station (NWS station code SIKA2) also has documented sub-hourly precipitation since 2005 (Diamond et al., 2013). Some variation in precipitation is observed at these two locations, but for the purposes of simplicity we train the statistical models using a single precipitation gauge, the PASI gauge at the Sitka Airport.

Through a combination of air photo interpretation and field mapping, the U.S. Forest Service has curated an inventory of more than 12,000 landslides in southeast Alaska, with records dating back to the early 20th century, known as the Tongass National



Forest Landslide Inventory (Tongass National Forest, 2017). To focus only on landslides likely to impact human safety and infrastructure, we subset the Tongass inventory to landslides within 2 km of the road network in Sitka, thus obtaining five days with recorded landslides out of 6,606 days with reported precipitation. We collected and synthesized information about the landslides near Sitka, including their timing and impacts.


**Table 1. Model naming system. We evaluated 136 models fit to the complete landslide inventory data and precipitation records. We also evaluated one frequentist logistic regression model fit to a subset of these data, reserving some data for validation (training-test), and one simpler alternative model based on historical landslide frequency.**

| Component | Type | Naming convention | Example |
|---|---|---|---|
| **Inference** | Frequentist | Model name begins with "F" | **F**L-1H1D |
| | Bayesian | Model name begins with "B" | **B**L-1H1D |
| **Model type** | Logistic regression | Second letter is "L" | F**L**-1H1D |
| | Poisson regression | Second letter is "P" | F**P**-1H1D |
| | Historical frequency | Labeled "simpler alternative" | SA |
| **Precipitation predictor 1** | Triggering precipitation (hours) | Number of hours is indicated as "#H" for 1, 3, 6, 12, or 24 hours | FL-**3H** |
| | Triggering precipitation (days) | Number of days is indicated as #D for 1,2,3,7, or 14 days; no hours indicated | FL-**1D** |
| **Precipitation predictor 2** | Antecedent precipitation | Number of days is indicated as "#D" for 1, 2, 3, 7, or 14 days | FL-1H**2D** |
| | No antecedent precipitation variable | No days indicated | FL-1H |
| **Training-test split (preferred model only)** | | | FL-TT-3H (frequentist logistic regression) |


### 2.2 Logistic and Poisson regression for estimating landslide hazard

Many previous works have used probabilistic techniques to predict landslide hazard (Brunetti et al., 2010; Berti et al., 2012; Tufano et al., 2019). In keeping with this practice, we used logistic regression to estimate the daily probability of landsliding as a function of precipitation. We also used Poisson regression to estimate intensity (number of landslides/day in the study region) as a proxy

for the magnitude of the event. The outputs of logistic and Poisson regressions are useful because they provide a nuanced understanding of relative landslide hazard that allows practitioners to identify multiple working thresholds that lead to different levels of community response.

Logistic and Poisson regression are generalized linear models that can include any number of predictor variables. To determine

the most effective precipitation timescale for estimating daily landslide hazard in Sitka, we tested a series of models with predictors at a range of timescales that include (a) triggering, or (b) triggering with antecedent precipitation. We considered two model set-



ups: the first (trigger-only) estimates daily landslide hazard (probability or intensity on day $d$) as a function of cumulative precipitation during a specified time period $t$, which is either a sub-daily interval of day $d$ or a time period leading up to and including day $d$. We investigated time periods $t$ of 1, 3, 6, 12, and 24 hours and 2, 3, 7, and 14 days. The second model set-up

(trigger-antecedent) introduces an additional predictor to describe cumulative precipitation during an antecedent time period $a$ preceding day $d$, and uses only sub-daily time periods for $t$. We considered antecedent periods of 1, 2, 3, 7, and 14 days.

For each day of recorded precipitation between 2002 and 2020 $d$ (6,606 days), we used a series of moving windows to extract the maximum cumulative precipitation in each sub-daily time period $t$ on that day and cumulative precipitation for all other time

periods $t$ and $a$ leading up to and including that day. This applies for both days with landslides and days without. For example, on a day with a landslide, 3-hour "triggering" precipitation ($P_t$) represents the highest cumulative three hours between midnight and 11:59 PM local time. A 1-day "antecedent" precipitation ($P_a$) period is the 24-hour period before midnight on the day of the landslide. The precipitation timescales we considered are designed to integrate with existing NWS precipitation forecasting products, which provide precipitation estimates for 3-hour intervals for the upcoming ~48 hours and 6-hour intervals for the

following ~48 hours.

The logistic regression models have the form:

$$
\begin{aligned}
y_d &\sim Bernoulli(p_d) \\
logit(p_d) &= \beta_0 + \beta_1 P_t + \beta_2 P_a
\end{aligned}
\tag{1}
$$

where $y_d$ is a binary indicator of whether a landslide was observed on day $d$, $p_d$ is the probability of having a landslide on day $d$,

and ~ indicates that $y_d$ is modeled as a Bernoulli distributed random variable. $\beta_0$ is the intercept of the generalized linear model; $\beta_1$ is the coefficient of cumulative precipitation ($P_t$) during time period $t$; and $\beta_2$ is the coefficient of antecedent precipitation ($P_a$) during time period $a$, which is excluded in the single-predictor models. Logistic regression models are indicated with an L in their name (Table 1).

The Poisson regression models have a similar form:

$$
\begin{aligned}
z_d &\sim Poisson(\lambda_d) \\
ln(\lambda_d) &= \alpha_0 + \alpha_1 P_t + \alpha_2 P_a
\end{aligned}
\tag{2}
$$


where $z_d$ is the number of landslides observed on day $d$, $\lambda_d$ is the average intensity of landsliding (landslides/day/area) on day $d$, $\alpha_0$ is the intercept, $\alpha_1$ is the coefficient of cumulative precipitation, and $\alpha_2$ is the coefficient of antecedent precipitation, again excluded in the single-predictor models. Poisson regression models are indicated with a P in their name (Table 1).

Landslide days (five days) are rare compared to non-landslide days (>6000), leading to an imbalance in the dataset that must be considered when setting decision thresholds.

We applied both frequentist (F, Table 1) and Bayesian (B, Table 1) approaches to fitting the logistic and Poisson regressions. Frequentist inference assumes that there is a true, unknown set of parameters and that the observed data result from an infinitely

repeatable sampling experiment. Frequentist 95% confidence intervals around the point estimate for a parameter have a 95% probability of including the true parameter value, if the experiment were repeated a large number of times. Bayesian inference, in contrast, provides posterior parameter estimates, which are probability distributions of all possible parameter estimates that are compatible with the observed landslide data and our prior knowledge, which is specified in the form of a probability distribution.



This is a useful property for estimating hazard from landslide inventories with few reported landslides because the posterior probability distribution quantifies how certain we can be of the parameter estimates, given few data points, and incorporates previous knowledge of landslide processes. A Bayesian 95% credibility interval contains 95% of the posterior probability, providing an arguably more intuitive estimate of uncertainty.

Both frequentist and Bayesian approaches have been applied in landslide research (e.g., Berti et al., 2012; Segoni et al., 2018). Frequentist approaches may be familiar to a wider range of users and are typically easy to apply out of the box in popular programming languages. Bayesian approaches offer potential advantages for small datasets, particularly because they quantify parameter uncertainty given the available data but are less commonly known and require sufficient expertise to select prior distributions. Here, we explore both inferences and compare their output and application.

For the Bayesian regressions, we chose weakly informative Student-$t$ prior distributions that reflect our expectations about landslide activity in Sitka, specifically that (1) landslide probability and the number of landslides should increase with increasing precipitation; (2) the probability of landsliding should be less than 50% at the mean precipitation in Sitka; (3) the average number of landslides should be fewer than 1 at the mean precipitation. Weakly informative Student-$t$ priors are recommended defaults for Bayesian logistic regression (Gelman et al., 2008) that encode prior knowledge without overly influencing regression results.
Additionally, when faced with an imbalanced dataset, as is the case here, such priors have been shown to produce stable regression coefficients, even in the case where there is near-perfect separation between landslide and non-landslide days (Gelman et al., 2008). Specifically, we chose the following:

$$\begin{aligned} \beta_0 &\sim Student - t(3, -3, 2.5) \\ \beta_1, \beta_2 &\sim Student - t(3, 3, 2.5) \\ \alpha_0 &\sim Student - t(3, 0.5, 1) \\ \alpha_1, \alpha_2 &\sim Student - t(3, -5, 1) \end{aligned} \qquad (3)$$

In the Bayesian regressions, precipitation values were standardized by subtracting the mean across all days and dividing by the
standard deviation, also known as a z-score. These priors refer to standardized data, where the intercepts $\beta_0$ and $\alpha_0$ indicate the expected values for probability and intensity at the mean precipitation value.

We fit the frequentist models using the R glm software package, which relies on iterative weighted least squares to estimate parameters (R Core Team, 2019). We fit the Bayesian models using the R brms software package version 2.17.0 (Bürkner, 2017),
which uses Hamiltonian Monte Carlo to estimate posterior parameter distributions as implemented in the Stan programming language (Stan Development Team, 2022). We ran four chains for 2000 iterations, discarding the first 500 draws as warm up. We checked the chains visually for convergence of parameter estimates; Gelman-Rubin convergence diagnostic (R-hat) values were in all cases 1, indicating convergence.

### 2.3 Model comparison and evaluation

We used several information criteria to compare models with different timescales of precipitation (1 hour to 14 days) to identify the best-performing model for use in a warning system. For the frequentist models, we calculated the Akaike Information Criterion (AIC) and Bayesian Information Criterion (BIC), which estimate out-of-sample prediction error. For the Bayesian models, we used approximate leave-one-out cross-validation as implemented in the R package loo version 2.5.1 (Vehtari et al., 2017), to estimate



out-of-sample predictive accuracy (Leave-One-Out Information Criterion, LOOIC). We then chose the respective logistic
regression and Poisson regression models with the lowest prediction error for further validation:

- FL-3H (Frequentist, logistic regression, 3-hour model),
- BL-3H (Bayesian, logistic regression, 3-hour model),
- FP-3H (Frequentist, Poisson, 3-hour model), and
- BP-3H (Bayesian, Poisson, 3-hour model).

Because the number of days with reported landslides is small compared to the number of days with no reported landslides, we
evaluated the sensitivity of these four selected models (FL-3H, BL-3H, FP-3H, and BP-3H) to individual landslide events using
leave-one-out cross validation. We removed each landslide event from the dataset and fit the models to the remaining data. We
then evaluated the difference in parameter estimates between the complete dataset and the leave-one-out dataset.

In addition to the leave-one-out exercise for models trained on the entire data series (minus one landslide day), we also split the
precipitation time series into a training and testing dataset to evaluate how well the model can be expected to predict landslides in
the future. The training dataset is composed of the precipitation and landslide records from November 2002 – November 2019.
The test dataset is from December 2019 – November 2020. We trained the logistic regression model on the training data and then
predicted the probability of landslides for all days between December 2019 – November 2020 based on observed precipitation
data. Although this "testing" window represents a relatively small portion of total days in the dataset, it does include 365 days and
two of five (40%) landslide days. To test the sensitivity of our results of the length of the training and test periods, we also flipped
the training and test periods (i.e., trained on the year 2020 and tested on the previous 17 years) and performed a similar validation.

One approach to understanding how well a model is able to predict landslides is to compare that model to alternative models (Table
1). We compared the skill of the best-fit frequentist and Bayesian logistic regression models (FL-3H, BL-3H) to a simpler
alternative model (SA) that randomly guesses whether a landslide will occur based on the historical daily frequency of landslides
(conceptually similar to tossing a weighted coin, where the sides are weighted according to how many landslide or non-landslide
days have been recorded). This also provides a way to evaluate the added value of applying a more complex statistical model over
a simpler model. The equation for this model is:

$$
\begin{aligned}
y_d &\sim Bernoulli(p_d) \\
p_d &= n_{ls}/n
\end{aligned}
\tag{4}
$$


where, as in the logistic regression models, $y_d$ indicates whether a landslide is observed, $p_d$ is the probability of landsliding, $n_{ls}$ is
the number of days with reported landslides, and $n$ is the total number of days on record.

We compared the best-fit logistic regression models (FL-3H, BL-3H) to the simpler model using the Brier Skill Score (BSS). The
Brier Score (BS) is the mean squared error of the predicted probabilities and is calculated as

$$
BS = \frac{\sum_{d=1}^{n}(p_d - y_d)^2}{n}
\tag{5}
$$

and where lower scores indicate better skill (Brier, 1950).

The BSS then compares the logistic regression model ($BS_{logistic}$) to the simpler alternative model ($BS_{SA}$):



$$BSS = 1 - \frac{BS_{logistic}}{BS_{SA}} \tag{6}$$

where a BSS of 0 indicates that the models have the same skill, a BSS > 0 indicates that the logistic regression model outperforms the reference historical daily frequency model (i.e., weighted coin toss), and a BSS < 0 indicates that the logistic regression model performs worse than the simpler model.

### 2.4 Setting multiple decision thresholds for different hazard levels

Although landslide probability and intensity can be estimated for any precipitation over a specified period using the fitted logistic and Poisson regression models, decision thresholds must be chosen to specify when to issue warnings. Extensive conversations with emergency responders and community leaders revealed a variety of perspectives and priorities for Sitka's landslide warning system and different levels of risk tolerance (Busch et al., 2021). For example, emergency responders who are concerned about the considerable cost of false alarms preferred a higher hazard threshold in favor of fewer false alarms. Other citizens were comfortable with some false alarms, preferring to be alerted whenever there was an elevated chance of landslides. These previous findings informed our selection of multiple warning levels because each threshold must compromise between missed and false alarms, and dual thresholds can inform different types of decision-making with different alert levels (e.g., Mirus et al., 2018b).

The trade-off between missed alarms and false alarms is illustrated using a confusion matrix. A confusion matrix for a single threshold is a 2 × 2 matrix that shows the number of true alarms, false alarms, missed alarms, and true no-alarms by comparing the predicted outcome based on the probabilistic model and threshold with the true outcome. Metrics calculated from the confusion matrix can reveal optimal thresholds based on the user's tolerance for false alarms or missed alarms. In imbalanced datasets with few landslide days and many no landslide days, typically applied metrics for logistic regression thresholding like accuracy and the Receiver-Operating-Characteristic (ROC) curve are less informative because they over-emphasize the importance of true no-alarms while masking the threshold's performance for true alarms and false alarms. For rare events, considering precision (ability to issue true alarms while avoiding false alarms) and recall (ability to issue true alarms while avoiding missed alarms) is preferable (Saito and Rehmsmeier, 2015).

Precision is defined as:

$$Precision = \frac{TrueAlarms}{TrueAlarms + FalseAlarms} \tag{7}$$

Recall is defined as:

$$Recall = \frac{TrueAlarms}{TrueAlarms + MissedAlarms} \tag{8}$$

To satisfy varying levels of risk tolerance within the community, we set two warning thresholds based on landslide probability estimated by best-performing frequentist logistic regression model (FL-3H). The lower threshold is set such that the system would have missed no landslide in the past (recall = 1), and the upper threshold is set such that every day with a landslide probability above the threshold has been associated with landslides in the past (precision = 1). Given the limited number of landslide days, a range of thresholds can achieve these results, calling for a heuristic approach in choosing final warning thresholds. We built a confusion matrix to illustrate how often warnings based on these thresholds would have been issued in the past and document the outcome of the event. We also used a confusion matrix to evaluate the number of warnings at each level that would have been





issued between December 2019 and 2020 using the model trained on an earlier section of the time series as described in section 2.3.

Ideal practice would include models tested with historical precipitation *forecasts* (rather than observed precipitation), but archived forecast data are not readily available. Instead, we set preliminary thresholds using observed precipitation totals, and necessarily
assume that forecasts are accurate. This introduces additional uncertainty in the landslide warning system, which relies on weather forecasts. Detailed analysis of the uncertainty in precipitation forecasts is beyond the scope of this paper, but validation and evaluation of the warning system could be used to refine warning thresholds.

**3 Results**

**3.1 Landslide events**

Over the 18 years with hourly precipitation records, five rain events in Sitka have resulted in one or more landslides (Fig. 2; Table 2). In most cases, landslide timing is known within the hour or can be estimated based on eyewitness constraints and the precipitation record. All five landslide events were characterized by a few hours of intense precipitation (Fig. 2).

Other landslides have occurred near Sitka but are > 2 km from the road network and sensitive infrastructure. For example, the
Starrigavan Landslide occurred several kilometers from town in 2014 and impacted a popular recreation area. Local accounts indicate that precipitation at the initiation site was much higher than precipitation observed at the Sitka Airport. Pronounced spatial heterogeneity in precipitation and weather is typical of southeast Alaska (Hennon et al. 2010; Shanley et al., 2015; Roth et al., 2018), which emphasizes the value of considering only very local (< 2 km from the road network) landslides for training prediction models using station-based precipitation data.


**Table 2. Summary statistics of recent landslide occurrences near Sitka. Timing is listed as "precise" when the landslide timing is known within 30 minutes, and "approximate" when landslide timing is known to a broader window (~12 hours) within the date of initiation. *Four landslides that occurred in fall 2020 are attributed to either the October 26 or November 1 storms, but it is not known which storm. The landslides that initiated < 2 km of the road network occurred on known dates.**

| Event name (local name for the most notable landslide) | Date, local time, precision | Total landslides | Landslides <2 km of road network | Failure type | Description of impacts and other notes |
|---|---|---|---|---|---|
| S. Kramer | Aug. 18, 2015, 9:30 am (precise) | 40+ | 6 | Debris flows initiated by shallow landslides | The debris flow near S. Kramer Avenue resulted in three fatalities. In addition to the six landslides counted here, multiple other landslides occurred in the region surrounding the study area. |
| Halibut Point Recreation Area | Sept. 4, 2017, Mid-day (approximate) | 1 | 1 | Landslide (type unspecified) | The landslide occurred in a recreation area, impacts unknown. |
| Medvejie | Sept. 20, 2019, 12:50 pm (precise) | 1 | 1 | Debris flow in an existing debris flow | Debris on the road to Medvejie Hatchery temporarily blocked access. |



| | | | | channel | |
|---|---|---|---|---|---|
| Harbor Mountain | Oct. 26, 2020<br><br>Early morning (approximate) | 6-10* | 2 | Debris flows | One debris flow temporarily blocked a highway. |
| Sand Dollar Drive | Nov. 1, 2020<br><br>6:00 pm (precise)<br><br>*and*<br><br>the night of Nov. 1 – Nov. 2<br><br>(approximate) | 2-6* | 2 | One debris flow; one fill-slope failure | The fill-slope failure occurred beneath a house on Sand Dollar Drive, impacting residential infrastructure. |


While antecedent precipitation conditions varied during these landslide events, the short-term (several hour) precipitation totals were high. For example, four of the events had peak 1-hour precipitation with >2-year return intervals as calculated by the NWS (Perica et al., 2012). Peak 3-hour precipitation during landslide events had between 2- and 25-year return intervals. Precise timing from eyewitness accounts and news records are available for the S. Kramer, Halibut Point, Medvejie, and Sand Dollar Dr.

landslides, which all occurred within a few hours of peak precipitation recorded at the Sitka Airport.

When compared to the record of all precipitation that did not initiate landslides over the last 18 years, the events that produce landslides occur at the extreme high end of the distribution of cumulative precipitation (Fig. 3).



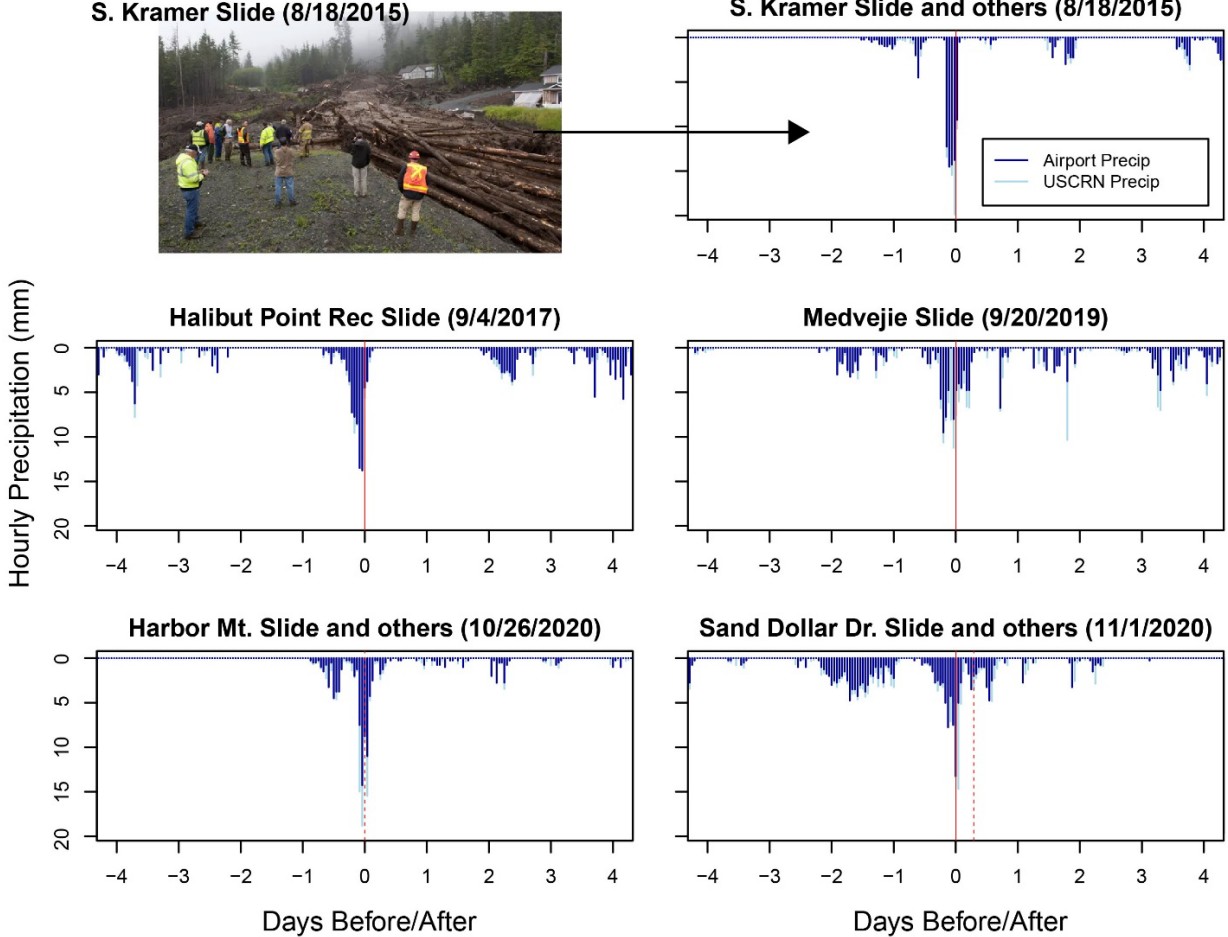

**Figure 2. Hourly precipitation before, during, and after landslide-initiating storms in Sitka. Landslide prediction models in this paper were trained on the longer record of hourly precipitation at the weather station at the Sitka Airport (darker blue bars). Recorded precipitation from a nearby weather station at the U.S. Climate Reference Network (USCRN) site in Sitka is also displayed for comparison. Landslide timing is indicated by a solid red line for events where timing is constrained to ~30 minutes and a dashed line where timing is constrained within ~12 hours. Photo courtesy of James Poulson, Sitka Sentinel.**

### 3.2 Landslide hazard prediction

We present results from both frequentist and Bayesian logistic regression and Poisson regression that predict landslide hazard (probability or intensity) based on precipitation totals over timescales ranging from 1 hour to 14 days (Fig. 3-6). Differences in model performance indicated by information criteria (section 3.3) show which timescales of precipitation provide the most useful prediction tools. For example, logistic regression based on 2-week precipitation totals is ineffective at separating landslide days from no-landslide days. Logistic regression based on 3-hour precipitation, however, does separate landslide days with some overlap (Fig. 3, 4).




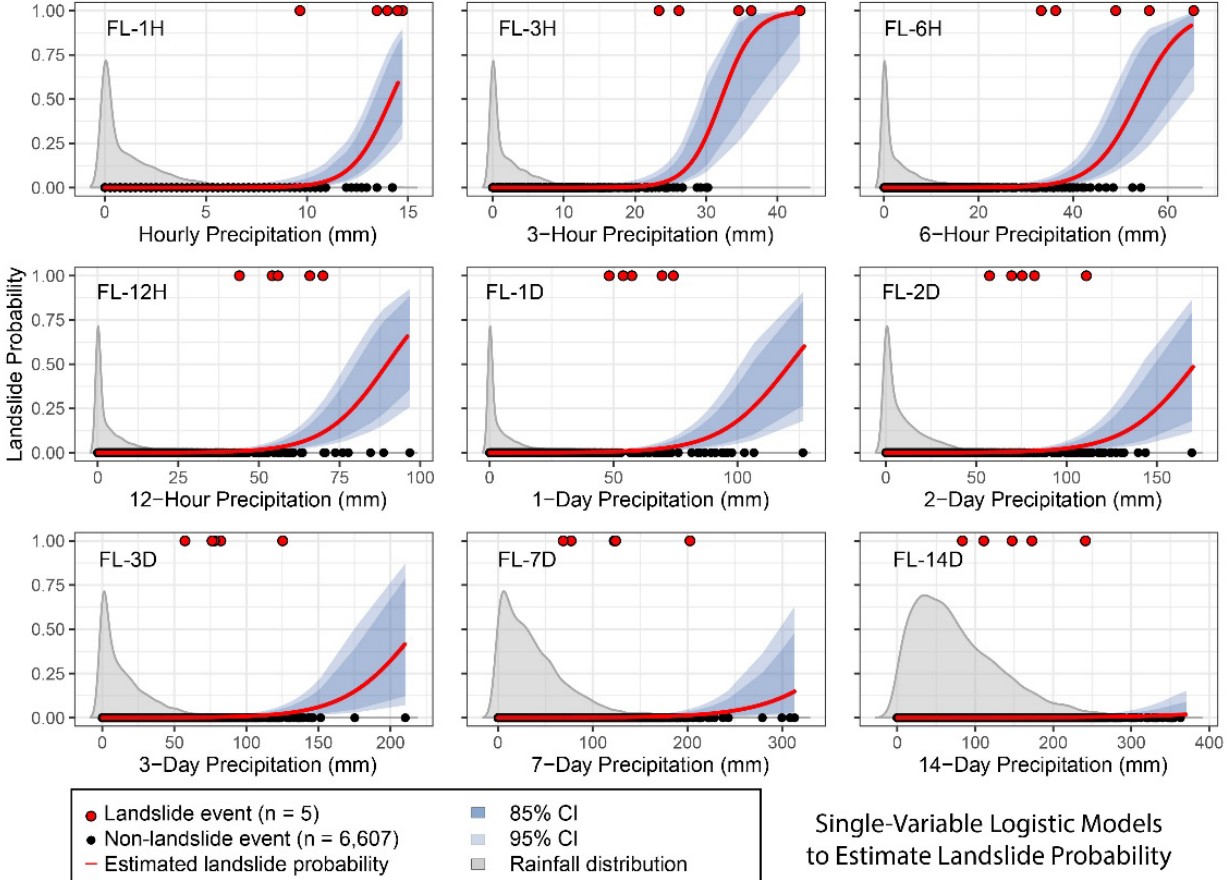

**Figure 3. Estimated daily landslide probability $p_d$ (red curve) from frequentist logistic regression based on different durations of precipitation from 1 hour to 2 weeks. Confidence intervals (CIs) were estimated based on standard error. Event outcomes (red circles) of 1 indicate at least one landslide occurred, while 0 indicates no landslide occurred. Three-hour precipitation produces the model with the best fit (lowest estimated out-of-sample prediction error based on Akaike Information Criterion (AIC) and Bayesian Information Criterion (BIC) fit parameters; Fig. 7). Kernel density distribution of all observed precipitation values are shown in gray, scaled for visual clarity.**





**Figure 4. Estimated daily landslide probability $p_d$ from Bayesian logistic regression, showing the posterior median (red curve) with 85% (darker purple) and 95% (lighter purple) Highest Density Intervals (HDIs). The 95% HDI is the posterior estimate of parameter uncertainty and contains 95% of the distribution of all parameter values compatible with the data and our prior knowledge. At a single precipitation value, the 95% HDI contains the true landslide probability with 95% probability, conditional on the data, the model, and our prior knowledge. Three-hour precipitation gives the best out-of-sample predictive accuracy as measured by leave-one-out Information Criterion (LOOIC) (Fig. 8).**



**Figure 5. Estimated daily average landslide count (red curve) from frequentist Poisson regression ($\lambda_d$) based on different durations of precipitation from 1 hour to 2 weeks. Event outcomes (red circles) of ≥1 indicate the number of landslides that were reported, while 0 indicates no landslide reported. Confidence intervals (CIs) were estimated based on standard error. Three-hour precipitation produces the model with the best fit (lowest estimated out-of-sample prediction error based on Akaike Information Criterion (AIC) and Bayesian Information Criterion (BIC) fit parameters (Fig. 7). Kernel density distribution of all observed precipitation values are shown in gray, scaled for visual clarity.**

420







**Figure 6. Posterior Bayesian Poisson regression results, showing the posterior median for the daily average estimated number of landslides $\lambda_d$ (red curve) with 85% (darker purple) and 95% (lighter purple) Highest Density Intervals (HDIs). Three-hour precipitation gives the best out-of-sample predictive accuracy as measured by leave-one-out Information Criterion (LOOIC) (Fig. 8).**

### 3.3 Model comparison

Models that incorporated short-term precipitation (three hours) demonstrated best fit to the data and lowest estimated out-of-sample prediction errors (Figs. 7–8). All models that incorporate a metric of precipitation on the day of the landslide show lower AIC, BIC, and LOOIC values than models based on accumulated precipitation over multiple days.





We note that for many of the Bayesian models with longer precipitation timescales (>1 day), Pareto-k values for some of the landslide days were > 0.7, indicating that these models would be very unlikely to predict a landslide at that precipitation value. In contrast, Pareto-k values for all landslide days in the 3-hour logistic regression model are < 0.7, confirming that this model is not overly sensitive to the individual landslide points.

**LOGISTIC REGRESSION**

|     | 1 | 2 | 3 | 7 | 14 | N/A |
|-----|-----|-----|-----|-----|-----|-----|
| N/A | 58.2 | 62.13 | 68.25 | 74.77 | 80.835 | |
| 1 | 27.89 | 27.66 | 27.82 | 27.13 | 28.08 | 26.08 |
| 3 | 24.23 | 21.81 | 23.11 | 22.83 | 24.45 | 22.82 |
| 6 | 31.99 | 29.02 | 31.32 | 31.79 | 32.92 | 30.98 |
| 12 | 50.15 | 48.86 | 49.99 | 49.76 | 50.42 | 48.46 |
| 24 | 60.17 | 59.40 | 60.11 | 59.67 | 60.04 | 58.2 |

AIC — Triggering (hrs) / Antecedent (days)

**POISSON**

|     | 1 | 2 | 3 | 7 | 14 | N/A |
|-----|-----|-----|-----|-----|-----|-----|
| N/A | 123.47 | 137.2 | 153.4 | 166.7 | 183.3 | |
| 1 | 41.90 | 41.78 | 42.21 | 39.39 | 42.30 | 40.33 |
| 3 | 40.29 | 35.77 | 37.66 | 38.42 | 40.89 | 40.02 |
| 6 | 49.62 | 43.85 | 48.25 | 48.83 | 51.48 | 49.87 |
| 12 | 106.4 | 105.0 | 106.9 | 105.5 | 106.4 | 105.0 |
| 24 | 124.5 | 125.0 | 125.4 | 124.5 | 124.2 | 123.5 |

Triggering (hrs) / Antecedent (days)

|     | 1 | 2 | 3 | 7 | 14 | N/A |
|-----|-----|-----|-----|-----|-----|-----|
| N/A | 71.79 | 75.72 | 81.84 | 88.36 | 94.42 | |
| 1 | 48.28 | 48.05 | 48.20 | 47.51 | 48.46 | 39.67 |
| 3 | 44.61 | 42.19 | 43.49 | 43.22 | 44.83 | 36.41 |
| 6 | 52.38 | 49.40 | 51.70 | 52.17 | 53.30 | 44.57 |
| 12 | 70.53 | 69.24 | 70.37 | 70.14 | 70.79 | 62.05 |
| 24 | 80.56 | 79.79 | 80.49 | 80.05 | 80.42 | 71.79 |

BIC — Triggering (hrs) / Antecedent (days)

|     | 1 | 2 | 3 | 7 | 14 | N/A |
|-----|-----|-----|-----|-----|-----|-----|
| N/A | 137.06 | 150.7 | 166.9 | 180.3 | 196.9 | |
| 1 | 62.29 | 62.17 | 62.59 | 59.78 | 62.68 | 53.92 |
| 3 | 60.68 | 56.15 | 58.05 | 58.80 | 61.27 | 53.61 |
| 6 | 70.01 | 64.23 | 68.64 | 69.21 | 71.86 | 63.46 |
| 12 | 126.8 | 125.4 | 127.3 | 125.8 | 126.7 | 118.61 |
| 24 | 145.8 | 145.4 | 145.8 | 144.8 | 144.6 | 137.06 |

Triggering (hrs) / Antecedent (days)

**Figure 7. Information criteria for a suite of frequentist models that estimate landslide probability (logistic regression) or number of landslides (Poisson regression) for a given precipitation characteristic. Each cell corresponds to a model with one or two precipitation parameters, including daily maximum cumulative precipitation measured over 1–24 hours ("triggering") and antecedent precipitation measured over 1–14 days. Lower Akaike Information Criterion (AIC)/ Bayesian Information Criterion (BIC) values (blue) correspond to better model fit and higher AIC/BIC values (red) correspond to worse model fit, with BIC more heavily penalizing complex models with multiple predictor variables. AIC and BIC scores are specific to a regression type and should not be compared between the logistic regression (probability output) and the Poisson regression (count output).**

**LOGISTIC REGRESSION**

|     | 1 | 2 | 3 | 7 | 14 | N/A |
|-----|-----|-----|-----|-----|-----|-----|
| N/A | 58.03 | 61.85 | 67.98 | 74.41 | 80.62 | |
| 1 | 27.87 | 29.31 | 29.53 | 28.88 | 29.72 | 26.17 |
| 3 | 23.43 | 23.70 | 24.04 | 24.06 | 25.03 | 22.12 |
| 6 | 30.72 | 30.72 | 32.50 | 33.07 | 34.09 | 30.68 |
| 12 | 50.46 | 51.04 | 51.98 | 51.55 | 51.50 | 48.86 |
| 24 | 60.41 | 60.58 | 61.56 | 60.32 | 60.58 | 58.03 |

LOOIC — Triggering (hrs) / Antecedent (days)

**POISSON**

|     | 1 | 2 | 3 | 7 | 14 | N/A |
|-----|-----|-----|-----|-----|-----|-----|
| N/A | 128.33 | 141.02 | 157.34 | 170.18 | 186.66 | |
| 1 | 46.22 | 52.84 | 54.53 | 42.71 | 48.29 | 43.19 |
| 3 | 41.46 | 39.48 | 41.65 | 43.62 | 45.73 | 40.32 |
| 6 | 51.93 | 49.41 | 52.64 | 55.75 | 59.17 | 50.21 |
| 12 | 112.8 | 114.82 | 122.39 | 113.65 | 118.55 | 110.05 |
| 24 | 133.25 | 137.14 | 140.08 | 133.23 | 137.34 | 128.33 |

Triggering (hrs) / Antecedent (days)

**Figure 8. Model comparison based on the Leave-One-Out-Information-Criterion (LOOIC) for a suite of Bayesian models that estimate landslide probability and average count for a given precipitation characteristic. Each cell corresponds to a model with one or two precipitation parameters, including daily maximum cumulative precipitation measured over 1–24 hours ("triggering") and antecedent precipitation measured over 1–14 days. Lower LOOIC values (blue) correspond to higher out-of-sample predictive accuracy, and higher LOOIC values (red) correspond to lower accuracy.**




We also qualitatively evaluated model fit by comparing the estimated landslide probability of the best-fit two-variable models that incorporate 3-hour triggering precipitation with 1-day (FL-3H1D) and 2-day (FL-3H2D) antecedent precipitation, respectively,

and model FL-3H the best-fit one-variable model that considers only 3-hour triggering precipitation (Fig. 9). Using either the 24- or 48-hour antecedent precipitation as another predictor does modify the probability contours in Fig. 9B and 9D, but the tradeoff between false versus failed alarms is largely unchanged across all threshold values. Most observed landslides cluster at high to extreme triggering precipitation values and low or moderate antecedent precipitation totals. Increased model complexity does not significantly improve model fit for the available database of landslide occurrence (Figs. 7–8).


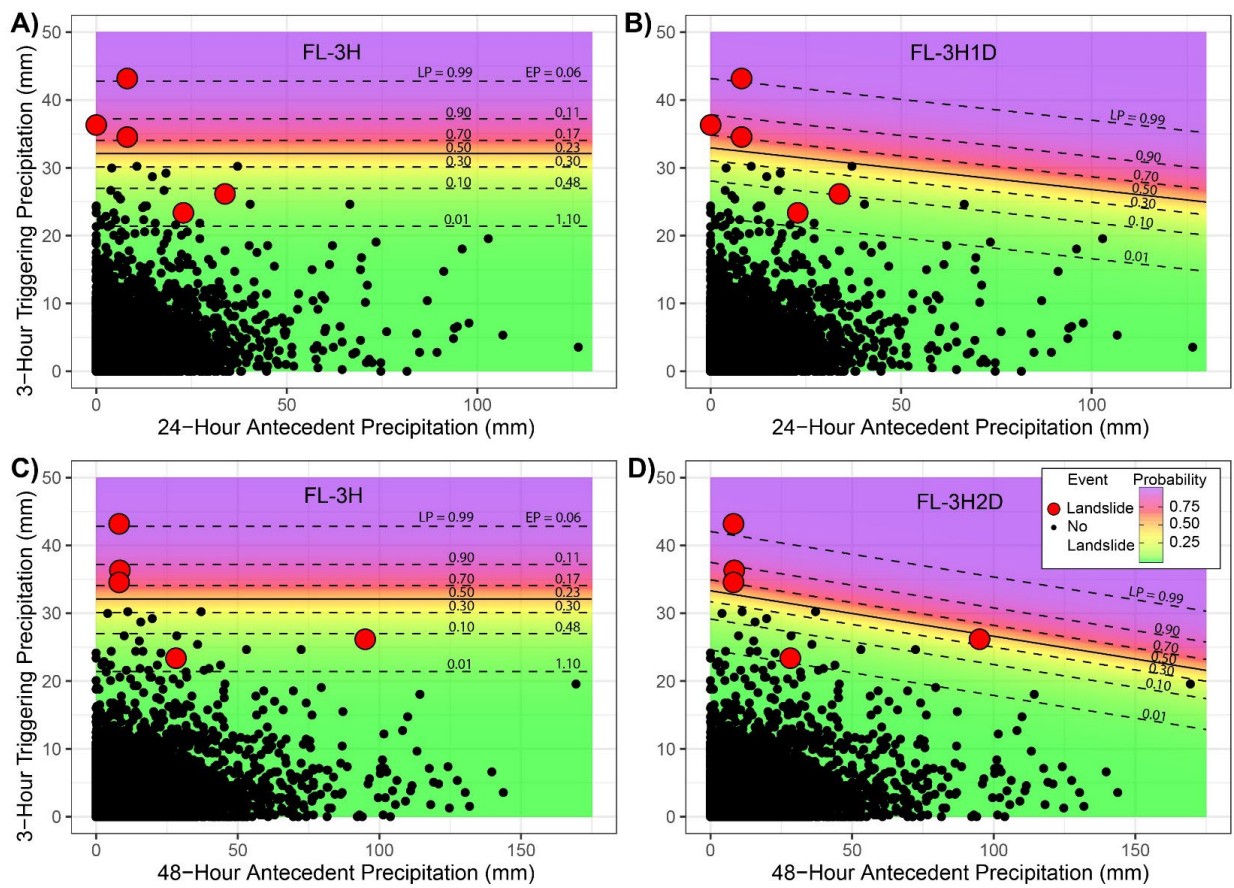

**Figure 9. Estimated landslide probability at varying precipitation values from the preferred "trigger-only" frequentist logistic regression model (FL-3H, panels A and C) compared to models that include antecedent precipitation (panels B and D, models FL-3H1D and FL-3H2D, respectively). The color gradient and black contours show estimated landslide probability with reported data shown as black**
**points (no landslide day) or red points (one or more landslides). Three-hour annual exceedance probabilities (EPs) reported by the NWS (Perica et al., 2012) also are plotted for the precipitation totals that generate the associated landslide probability (LP) contours from frequentist logistic regression in panels A and C.**





**Figure 10. Leave-one-out cross validation for the preferred frequentist logistic regression model FL-3H (left column) and Poisson 3-hour model FP-3H (right column). Similar to Fig. 3 and Fig. 5, solid red points are landslide events, black points are non-landslide events, red lines show model estimates, and the dark and light shaded regions show 85% and 95% confidence intervals, respectively. Hollow red circles and dashed black lines show the landslide event that was omitted from each run. Model coefficient estimates are shown in the bottom panel with 95% confidence intervals based on the standard error. Model output and coefficient estimates remain largely unchanged when an individual landslide event is "missed" in the inventory, but the uncertainty bounds of the logistic regression and Poisson regression are sensitive to "missing" the landslide events with the lowest and highest precipitation, respectively.**





Given the small number of observed landslide events in the dataset, we evaluated the sensitivity of the 3-hour models to individual landslide events using leave-one-out cross validation (Fig. 10 and Supplementary Fig. S1). Parameter estimates and their 95% confidence intervals for the leave-one-out models and the full dataset logistic regression models (FL-3H) overlap, indicating no

relevant difference (Fig. 10). That we cannot distinguish these model parameters demonstrates that the model is not particularly sensitive to individual landslide points. The confidence intervals of the parameter estimates for the Poisson models also overlap with each other, with very high similarity in most cases, but we observe that the model is particularly sensitive to the single landslide day with six individual landslides. Further evaluations focus on the frequentist model (for ease of implementation) with lowest prediction error: frequentist model FL-3H.

**3.4 Thresholds and predictive performance**

Based on predicted daily landslide probability from the preferred logistic regression models (FL-3H, BL-3H), we established two decision thresholds for a landslide warning system (Fig. 11). A lower threshold was set at a probability of 0.01; in the past, this threshold would have resulted in no missed alarms (recall = 1). Any threshold below a probability of 0.023, based on model FL-3H estimates, results in a recall = 1; the threshold that maximizes precision at a recall = 1 would be 0.023, resulting in a precision

of 0.22. We took a conservative approach and set the threshold lower, at 0.01. At 0.01, precision = 0.15, indicating that 28 false alarms would have occurred between 2002 and 2020 if this threshold had been used in the past.

Frequentist logistic regression indicates that a probability of 0.01 corresponds to a precipitation total of 21.3 mm in 3 hours (0.84 inch). Bayesian logistic regression indicates that a probability of 0.01 could correspond to precipitation values between 17.4 and

24.0 mm (95% HDI) (0.685 to 0.945 inch). An upper threshold that would have resulted in no false alarms (precision = 1) was set at 0.70. Based on model FL-3H estimates, any threshold above a probability of 0.31 results in a precision of 1, and thresholds that maximize recall for a precision of 1 range from 0.31 to 0.74 (Fig. 11). This wide range results from both few reported landslides and few reported precipitation values at the tail end of the precipitation distribution. At 0.70, recall = 0.6, indicating that two of the five reported landslide events occurred below this threshold. At this probability, frequentist logistic regression corresponds to

precipitation of 34.0 mm in 3 hours (1.34 inches) and Bayesian logistic regression indicates precipitation between 31.0 and 39.2 mm (95% HDI) (1.22 to 1.54 inch).

At a precipitation total of 21.3 mm in 3 hours, the 3-hour Poisson models predict the occurrence of 0.015 landslides per day on average in the study area (FP-3H) or between 0.0034 and 0.031 landslides (95% HDI, BP-3H). At 34.0 mm of precipitation in 3

hours, Poisson regression predicts 0.56 (FP-3H) or between 0.25 and 0.86 landslides (95% HDI, BP-3H).

Decision thresholds for the landslide early warning system in Sitka were based on consideration of these ranges and our judgement. Using the frequentist logistic regression (FL-3H) model, probabilities < 0.01 were considered "Low" hazard; 0.01 < 0.70 were considered "Moderate" hazard; and > 0.70 were considered "High" hazard. The confusion matrix in Table 3 shows the warning

levels that would have been generated by observed precipitation in 2002–2020 and the actual outcome. Probability of landslides for exceedance of our two alerts are consistent with other areas where dual- or multiple-thresholds are used (e.g., Chleborad et al., 2006).



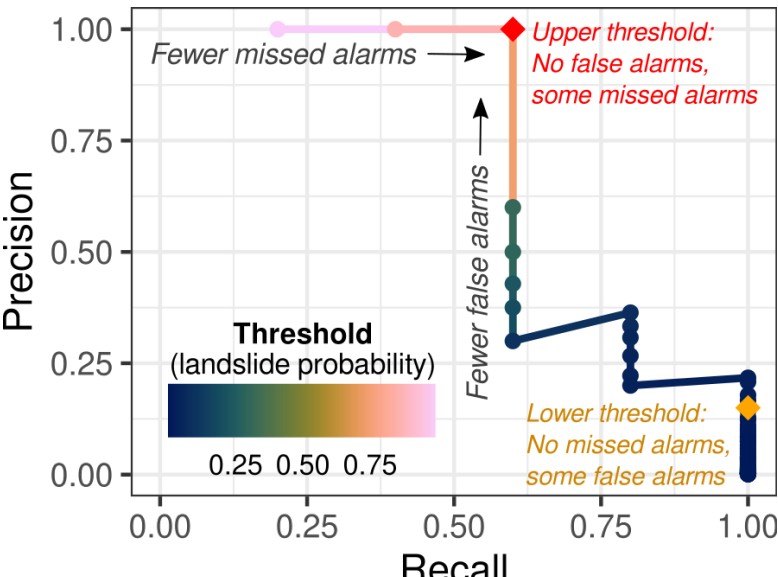

**Figure 11. Precision-Recall curve based the preferred frequentist logistic regression 3-hour model (FL-3H). Recall is the true alarm rate and precision is the rate of true alarms/true alarms + false alarms. The upper threshold (red diamond) was heuristically set at a daily landslide probability of 0.7 to result in no false alarms (precision = 1). The lower threshold (orange diamond) was set at a probability of 0.1 to result in no missed alarms (recall = 1).**

**Table 3: Warning levels that would have been generated between 2002 and 2020 by model FL-3H and the selected decision thresholds, showing the number of times each warning level would have been reached and the actual outcome. For example, a "High" warning would have been reached three times, and landslides occurred all three times; similarly, zero landslides occurred during times when "Low" probability of landslides would have been predicted by the model.**

|  | "Low" Warning | "Moderate" Warning | "High" Warning |
|---|---|---|---|
| Landslide | 0 | 2 | 3 |
| No landslide | 6573 | 28 | 0 |

At these thresholds, a moderate warning level would have been issued on 28 days between 2002 and 2020; two of those days actually resulted in landslides (Fig. 12A). A high warning level would have been issued three times, with all three days actually resulting in landslides. No landslide warning would have been issued on 6573 days, and no landslides would have occurred on a day without a landslide warning. This is useful for estimating the impact on a community based on the frequency of warnings. For example, the moderate warning level would have been issued 1-2 times per year (on average) in the historical record. However, while the confusion matrix summarizes how the model would have behaved in the past, it is not an indicator of how well the model can predict landslides because it uses the same dataset for validation as was used for training.

In the frequentist logistic regression model FL-TT-3H, we split the precipitation time series into training (November 12, 2002 – November 30, 2019) and test data (December 1, 2019 – November 30, 2020). Model FL-TT-3H is trained using only three landslide days and 6225 non-landslide days, and with thresholds at 0.01 and 0.70. This version of the model predicts elevated landslide probabilities on the days when landslides occurred in October and November of 2020 (Fig. 12B). Table 4 presents a confusion matrix for predicted warning levels for all days in the test dataset. A moderate warning would have been issued on two days, and





both of those days did see landslides. No high warnings would have been issued. A low warning level would have been present on the remaining 363 days, with no landslides occurring. When we flip the training and testing periods and apply the same thresholds (probability = 0.01 and 0.7), the model would have issued warnings for all three testing landslide events, corresponding to a recall
of 1 and no missed alarms (Supplementary Fig. S2, Table S1).

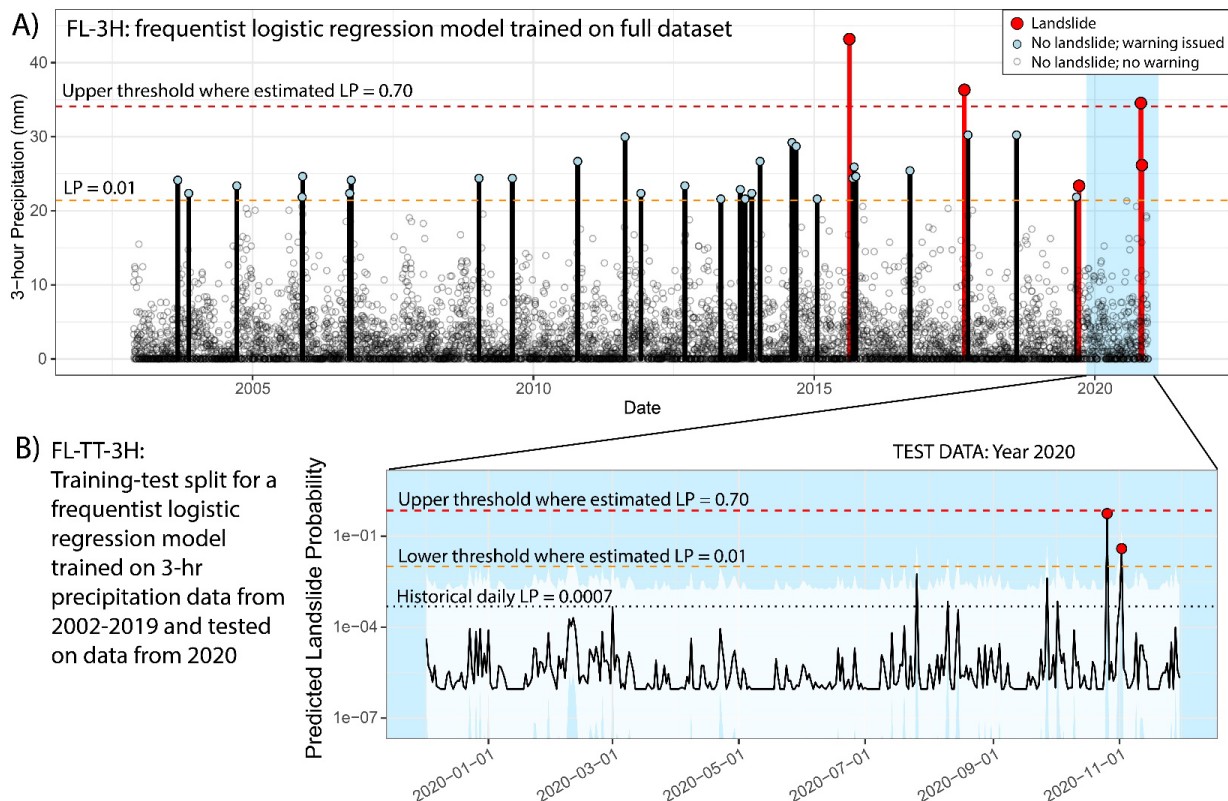

**Figure 12. (A) Timing of "moderate hazard" storm events (probability > 0.01 in the 3-hour frequentist logistic regression model, FL-3H) within the full period of record. All five landslide events in Sitka occurred within the last several years of record. Landslide probability**
**is abbreviated as LP. (B) "Test" data for a model FL-TT-3H trained on the time series from 2002-2019. The black line indicates the point estimate and the gray field shows the 95% standard error. The model correctly predicts elevated hazard during the two landslide-initiating storms in 2020. The thresholds where probability = 0.01 and P = 0.70 are similar but not exactly the same in the two models (FL-3H and FL-TT-3H), such that one of the 2020 storms would have been predicted as "High" hazard (P > 0.70) in model FL-3H trained on the full dataset (A) but "Moderate" (0.01 < probability < 0.70) hazard in model FL-TT-3H trained on a subset of the dataset (B).**


**Table 4: Confusion matrix for 2020 predictions, based on model FL-TT-3H trained on 2002–2019 and with thresholds at probabilities of 0.01 and 0.7, showing the number of times each warning level would have been reached and the actual outcome. For example, a "Moderate" warning would have been reached twice, and landslides occurred both times.**

|  | Low | Moderate | High |
|---|---|---|---|
| Landslide | 0 | 2 | 0 |
| No landslide | 363 | 0 | 0 |





We also compared the performance of the logistic regression model FL-3H to a simpler model that randomly guesses if a landslide will occur based on historical frequency. The logistic regression model FL-TT-3H far outperforms random guessing (BSS = 0.44) for December 2019 – November 2020.

**4 Discussion**

In this study, we applied logistic regression and Poisson regression to develop probabilistic daily estimates of landslide hazard in
Sitka, Alaska, using limited landslide inventory data and nearly 20 years of hourly precipitation records (2002–2020). Based on these hazard estimates, we established two decision thresholds for landslide warning for implementation in a public-facing online dashboard that is driven with NWS forecast data and is automatically updated in real-time.

**4.1 Probability based decision thresholds for landslide warning**

Most commonly applied approaches to determining thresholds for landslide initiation seek to distinguish between precipitation
and/or hydrological conditions that lead to landsliding from those that do not, or to determine a boundary below which landslides have not been previously observed. A disadvantage to such thresholds is that they (by design) provide only a binary outcome and no estimate of relative hazard. Probabilistic models, in contrast, estimate hazard and its uncertainty at every value of a predictor variable (e.g., maximum daily 3-hour precipitation), providing richer information than a binary threshold.

Identifying the most appropriate timescales for triggering and antecedent precipitation data influences the accuracy of landslide prediction tools (Gariano et al., 2020). By exploring the fit and predictive performance of selected precipitation timescales, including both triggering and antecedent precipitation, our models also provide insight into the physical processes that govern landslide initiation near Sitka.

We found that the 3-hour precipitation predictors best fit the data (e.g., models FL-3H, FP-3H, BL-3H, BP-3H), with negligible improvement in models that further incorporate 24- or 48-hour antecedent precipitation (e.g., models FL-3H1D, FL-3H2D, BL-3H1D). Including antecedent precipitation over timescales >48 hours reduced model fit. Compared to some examples of cumulative precipitation thresholds in Seattle, Washington, which incorporate 3-day and 15-day antecedent precipitation totals (Chleborad et al., 2006; Scheevel et al., 2017), these timescales in Sitka are short. Similarly, intensity duration thresholds in Seattle rely on
additional information on antecedent wetness for accurate performance (Godt et al., 2006).

These short timescales and lack of improvement with antecedent information for Sitka may result from multiple factors, including the steep topography, thin and locally permeable colluvial soils (Swanston and Marion, 1991), preferential flow and fracture-driven hydrology, unconstrained meso-scale storm patterns associated with landslide initiation in Sitka, and the small number of observed
landslides in the dataset. We hypothesize that the importance of relatively short periods of intense precipitation in Sitka reflects the rapid hydrologic response of shallow, porous soils on fractured bedrock that commonly are near saturation at critical failure depths. Antecedent information may be less predictive in this environment than in regions with thick or impermeable soils. Conversely, the lower performance of models using the 1-hour timescale indicates that shorter-duration bursts of intense rainfall are not necessarily sufficient to trigger landslides, and that some degree of sustained infiltration of rainfall is still needed.






Previous investigations on Chichagof Island, north of Sitka, demonstrate a rapid hydrologic response, with peaks in shallow pore pressure occurring within a few hours of observed precipitation and dissipating within several hours (Sidle, 1984); one investigation found that a shallow debris slide was most likely associated with maximum short-term intensity (2–6 hours) of precipitation, rather than storm totals (Sidle and Swanston, 1981).


Although the probabilistic outputs of logistic regression and Poisson regression are useful for understanding the relative magnitude of landslide hazard, it was necessary to establish decision boundaries for warning levels to communicate hazard to the public. As described in section 3.5, we selected two decision boundaries where frequentist logistic regression of maximum 3-hour precipitation (FL-3H) estimates a daily landslide probability of 0.01 and 0.70 for "Moderate" and "High" warning levels,

respectively. Based on the historical record, moderate warnings would have been generated 31 times since 2002, and correctly predicted landslides only 3 times (Fig. 12). This means that there are many false alarms at the moderate warning level, but, by design, no missed alarms for this lower threshold. In comparison, the higher threshold was only crossed three times since 2002, all of which resulted in >1 landslide. These outcomes demonstrate the utility of having a tiered warning system, which provides more nuanced information about landslide hazard during a forecasted storm. As described below, these estimates do not account for

unquantified error in precipitation forecasts or meso-scale atmospheric processes, which can generate above-threshold precipitation that may not be captured with traditional rain gauge networks (Collins et al., 2020).

The statistical models presented here are designed to be adaptable as additional data and observation allow us to validate and refine the models. Bayesian reasoning in particular acknowledges such updates by evaluating how much has been learned in the revised

posterior. Hydrologic monitoring has recently been implemented in Sitka, but the available data record is relatively short. Further evaluation of this hydrologic time series could improve understanding of the hydrologic conditions that trigger landslides. For example, the preferred statistical models could be updated for seamless integration of additional hydrologic data or other predictor variables into the models if they improve prediction accuracy.

### 4.2 Few landslide observations and many no-landslide observations: strengths

While the overall dataset of precipitation observations is large (>6,600 days of hourly precipitation record), the number of landslide-inducing events in this highly localized dataset is small (five landslide events <2 km from the road network). This imbalanced dataset results in large model uncertainty for extremely high precipitation values that have been rarely observed.

Our work confirms that useful precipitation thresholds may be defined without landslide events by including the distribution of

non-triggering events, as demonstrated in recent investigations (Peres and Cancelliere, 2021). This is possible because high data availability for non-triggering events does constrain the relatively low probability of landslides at low precipitation values. In other words, prediction models built with non-triggering events provide larger datasets than those that consider only landslide-triggering events and can be robust when considered alone or in combination with known landslide occurrence. The value of low precipitation totals and non-triggering events are often overlooked in landslide prediction studies, but the large number of observations results

in statistically robust models. This is reflected in our leave-one-out cross validation results, where we show that logistic regression parameter estimates are insensitive to individual landslide events, and our training-test thresholding results, where we show that a model trained with only three landslide events is able to issue warnings during two test events. Poisson regression results are more sensitive to the largest landslide event, but a model trained without the largest landslide event would have predicted the occurrence



of multiple landslides. We find that our preferred logistic regression models are more skillful in estimating landslide hazard than
a simpler alternative model in which daily landslide probability is estimated by historical daily frequency, analogous to randomly
guessing whether a landslide will occur based on how often they have occurred in the past.

Based on previous surveys, conversations, and feedback with the community in Sitka (Busch et al., 2021; Izenberg et al., 2022),
one particularly valuable result of our modeling is a well-constrained "low" probability of landsliding, which is possible due to the
extensive non-triggering events in the precipitation record. Identifying the times when landslide occurrence is *not* likely (i.e., <1%
daily probability) allows residents to manage anxiety while living in a potentially hazardous landscape with frequent and sustained
rainfall throughout the year.

**4.3 Few landslide observations and many no-landslide observations: challenges**

Although few landslide events combined with many non-landslide days resulted in robust statistical models, setting and validating
decision thresholds based on only five landslide days presented additional challenges. For example, although Receiver-Operating-
Characteristic (ROC) curves are commonly used to select optimal decision thresholds based on logistic regression (Giannecchini
et al., 2016), this approach is not as informative for highly imbalanced datasets as a Precision-Recall curve (Saito and Rehmsmeier,
2015) because a wide range of thresholds give a low false alarm rate. We therefore opted to consider the Precision-Recall curve
(Fig. 11), which provides more information about how well a threshold can distinguish between true alarms and false alarms.

However, even when using the more appropriate Precision-Recall curve, three sources of uncertainty make the choice of threshold
challenging: (1) because there are only five landslide events, a range of decision thresholds result in identical levels of precision
and recall. For the upper threshold, for example, threshold values with the same precision and recall range from daily landslide
probabilities of 0.31 to 0.74 based on the frequentist 3-hour logistic regression model (FL-3H), (2) at any of these potential
threshold levels, the given probability could be associated with a range of precipitation values; for an upper threshold of 0.70, for
example, these range from 31.0 to 39.2 mm with 95% posterior probability, based on the Bayesian 3-hour logistic regression model
(BL-3H), and (3) without further correction, logistic regression based on imbalanced datasets can underestimate landslide
probability (King and Zeng, 2003). A conservative yet easy-to-implement approach is to reduce the lower threshold below an
optimal balance of precision and recall (threshold tuning), particularly if the primary goal of the (lower) decision threshold is to
reduce risk of missed alarms.

Despite these uncertainties, two decision thresholds corresponding to specific precipitation values were required for
implementation in the warning system to integrate with weather forecasts. We therefore chose a heuristic approach based on expert
judgement to select precipitation values within the range of thresholds that lead to the desired precision and recall. We set the lower
threshold to a probability of 0.01 in model FL-3H, which is generated by 21.3 mm in 3 hours (0.84 inch). We set the upper threshold
at a probability of 0.70 in model FL-3H, which is generated by 34.0 mm in 3 hours (1.34 inches).

We also found challenges associated with the timing of landslides over the 18-year record: although the hourly precipitation record
in Sitka starts in 2002, no landslides with well-constrained timing were reported until 2015. This presents an additional challenge
when validating the selected thresholds for future predictive performance. An ideal approach would be to iteratively split the
dataset into multiple training and test groups (k-fold cross-validation), each time fitting statistical models to the training set and



testing performance with the test set. Because landslides in this dataset only occur in the final third of the dataset, k-fold cross-validation results in many training sets with no reported landslides, which are then unable to predict elevated landslide hazard and are not useful estimates of performance because the models applied in the warning system do include reported landslides in the

training data. We emphasize that in the training-test split presented here (FL-TT-3H), a model trained on only three landslide points with our selected probability thresholds is able to issue moderate warnings during both testing landslide events. Although no false alarms occurred during the testing period, some false alarms can be expected in the future, as the low warning threshold has been exceeded in the past without triggering landslides (Table 3). More reported landslides in the Sitka area in the future would allow for more extensive validation of the thresholds. We consider several potential explanations for this inconsistent frequency within

the study period.

First, it is possible that small, isolated slope failures may have occurred prior to the major event in 2015 but were not well documented. We consider it unlikely, however, that large or extensive landslides occurred and were not observed between 2002 and 2015 in Sitka. A second potential explanation is that, although "moderate" hazard rain events have occurred throughout the

period of record (Fig. 12), the only "high" hazard events on record occurred after 2015. Global warming is predicted to result in increased frequency of extreme precipitation in upcoming years and decades, but further study is needed to evaluate the links between changing precipitation patterns and landslide occurrence in southeast Alaska. A third potential explanation is increased human alteration of hillslopes. At least one recent landslide occurred in human-made fill material, which may have increased susceptibility to landslide failure compared to a natural slope by altering hydrologic characteristics and slope strength through

vegetation removal, slope cutting, and addition of fill material (e.g., Beville et al., 2010; Bozzolan et al., 2020; Johnston et al., 2021). If intense precipitation events are becoming more frequent or if slope modification increases through expanded urban development (or both), landslides in the study area are likely to become more frequent in the future.

### 4.4 Experience with frequentist and Bayesian inference for estimating landslide hazard

We explored both frequentist and Bayesian approaches to fitting logistic regression and Poisson regression models for estimating

landslide hazard. By design, both of these approaches produced similar results; however, they do have different implications for use and interpretation. Frequentist inference remains more commonly used in landslide research (Melillo et al., 2018; Segoni et al., 2018), indicating familiarity, and frequentist approaches tend to be straightforward to implement in commonly used statistical modeling software, like R glm applied here. However, when considering imbalanced datasets with rare events, frequentist logistic regression may underestimate landslide probability (King and Zeng, 2003), and parameter estimates may be unstable when near

perfect separation between landslide and no landslide days occurs, as is the case in this dataset. We note that statistical strategies exist to correct for underestimation (King and Zeng, 2003) and to obtain stable parameter estimates (Kosmidis and Firth, 2021), which could be applied if Bayesian inference were unavailable as a cross-check. Additionally, frequentist confidence intervals must be estimated in an additional step, and their interpretation is arguably less intuitive than Bayesian credibility intervals. However, in this case we found frequentist inference to still be useful for defining heuristic decision thresholds.


Bayesian inference remains less common in landslide research, and although additional expertise is required to set prior distributions and interpret the results (e.g., McElreath, 2020; Bürkner, 2017) Bayesian inference allows for incorporation of prior knowledge, which is advantageous when few landslide events are reported. Here, we encoded our prior knowledge that landslide activity is likely to increase with increasing precipitation in a weakly informative prior, which by design has only a small influence





on the posterior distribution. When few data are available, more informative priors based on other studies could be used to, for example, tell the model about a distribution of outcomes that are known to be possible from nearby areas, but were not observed in the small dataset at hand. Weakly informative priors have also been shown to lead to stable parameter estimates in the case of imbalanced datasets with rare events, which overcomes the problem of unstable parameter estimates that frequentist logistic regression can show without an additional correction, making these Bayesian models better suited to estimating hazard from

imbalanced datasets (Gelman et al., 2008). Posterior distributions of parameter estimates provide intrinsic estimates of uncertainty learned from the data, which informed our understanding of the range of precipitation values that could be associated with a given decision threshold.

Overall, we conclude that frequentist models are familiar and easy to implement, but Bayesian models capture the rare-events

problem more explicitly and allow for better understanding of uncertainty. Either model can be effectively used for probabilistic landslide models and to determine meaningful decision thresholds. Here we present thresholds for the easy-to-implement frequentist model, but consideration of the best-fit Bayesian model and parameter uncertainty improved our understanding of both models' strengths and weaknesses. Furthermore, either workflow is transferrable to other regions, but they would need to be trained on local data.

**4.5 Landslide prediction and uncertainty based on weather forecasts**

Accurate precipitation and landslide timing data facilitated the development of robust thresholds for low, moderate, and high landslide potential. Implementation of these thresholds into actionable information to provide *advance* warning of landslide potential hinges upon accurate precipitation forecasts. Uncertainty in the forecasted precipitation is added to uncertainty of the model and decision threshold. As storms approach and precipitation forecasts become more constrained, the precipitation

uncertainty will be reduced (Fabry and Seed, 2009; Ashok and Pekkat, 2022). Thus, landslide predictions for the future (days in advance) become more accurate as the storm approaches (hours in advance). Effective warning education can encourage residents to stay alert for updated landslide predictions. Further studies would be useful to better quantify the magnitude of error expected in dynamic storm forecasting and the relationship between forecast uncertainty and time into the future.

Models trained on and applied to precipitation data from a single monitoring station (Sitka Airport) cannot account for spatial variability in precipitation totals. Although the geographically small study area described here is intended to minimize these impacts, mountainous areas (like Sitka) are characterized by spatially variable climate and weather patterns (Johnson and Hanson, 1995; Tullos et al., 2016; Napoli et al., 2019). Meso-scale atmospheric processes linked to spatial distribution of landslide initiation are difficult to model (Collins et al., 2020) and not typically incorporated in kilometers-scale precipitation forecasts.


Predicted landslide hazard could also be complemented by applying the model to recent precipitation observations as an estimate of current hazard. "Nowcasting" by looking at observed precipitation (e.g., Kirschbaum and Stanley, 2018) incorporates instrument and model error, but not weather forecast error, and can alert residents to hazardous conditions that exceeded previous predictions. This type of information provides indicators of immediate hazard but is less useful for developing emergency response plans or

informing operational decisions, which require sufficient lead time to take suitable action. In Sitka, for example, the observed landslides with high resolution timing data occurred 1–3 hours following peak precipitation, which may still provide valuable time for emergency responders and risk-averse individuals to take actions that reduce risk if precipitation totals exceed forecasts.



**4.6 Application to landslide early warning system in Sitka, Alaska**

In Sitka, our best-fit frequentist model FL-3H (based on 3-hour precipitation) with the three warning levels (low, moderate, high)
described in section 3.5 has been applied to a public-facing dashboard for situational awareness. This dashboard provides residents, emergency planners, and NWS forecasters with near-real-time updates of current and predicted landslide hazard (referred to as "risk" in the dashboard for ease of use by a non-technical audience) and suggests actions to mitigate risk. In Sitka, individual differences in risk tolerance create a need for contextualized risk information to be available to everyone in the community (Busch et al., 2021). To provide this service, project members worked with the community, web developers, and NWS forecasters to
construct a series of warning levels that indicate the three levels of landslide hazard developed and tested in this work. The beta version of this dashboard is accessible at sitkalandslide.org, which at the time of writing, is functionally serving as a landslide early warning system used by the public to inform individual decision making and by NWS forecasters to guide special weather watch, warning, and advisory products.

**5 Conclusions**

In this study we developed and evaluated probabilistic models for landslide hazard estimation built with a small landslide inventory. The best-fit models used 3-hour triggering precipitation only. Including antecedent precipitation in addition to triggering rainfall did not improve model fit for the available database of landslide occurrence relative to using only the triggering precipitation conditions.

Despite the small number of landslide events (five days with landslides), a large dataset of non-triggering events produces robust model results, albeit with higher uncertainty at high precipitation values. Validation through leave-one-out analysis demonstrates that the model is robust even if we assume that we missed a landslide event. Furthermore, training the model on only three of five landslide events and thousands of no-landslide events would still have resulted in a model that could correctly predict the subsequent two landslide events. This model outperforms a much simpler alternative model based on historical landslide frequency.
Combined with probabilistic models, the small number of landslide events allowed for the development of usable decision thresholds for landslide warning.

Although frequentist and Bayesian inference produce similar estimates of landslide hazard by design, they do have different implications for use and interpretation: frequentist models are familiar and easy to implement, but Bayesian models capture the
rare-events problem more explicitly and allow for better understanding of uncertainty.

Developing precipitation thresholds based on time intervals (e.g., 3 hours) that match NWS forecasting products allows for application to landslide predictions within the NWS operational framework. This landslide early warning system was developed in partnership with the community and prioritized community needs identified in previous studies (Busch et al., 2021). A publicly
accessible web dashboard, sitkalandslide.org, uses our preferred frequentist logistic regression model (FL-3H) and precipitation thresholds to display current landslide hazard (based on recent precipitation) and "forecasted" landslide hazard (based on NWS forecasts) in real time.

**Acknowledgements**



This research was supported by National Science Foundation funding (Award #1831770). Lisa Luna was supported by the Deutsche Forschungsgemeinschaft (DFG) research training group "Natural Hazards and Risks in a Changing World" (NatRiskChange GRK 2043). We would also like to thank our collaborators including Robert Lempert, Ryan Brown, and Max Izenberg (RAND Corporation) and Lisa Busch (Sitka Sound Science Center) for leading conversations with Sitkans about risk perception; Jacquie Foss (U.S. Forest Service) for curating and clarifying entries in the Tongass Landslide Inventory and documenting landslide impacts; Jeff Frankl, Klaas Hoekema and Steph Wall (Azavea) for designing and building the public-facing warning dashboard; and Cora Siebert and Jacyn Schmidt (Sitka Sound Science Center) for documenting storm impacts. We thank Corina Cerovski-Darriau (USGS) and Ugur Özturk (University of Potsdam) for their thoughtful reviews, which allowed us to improve the clarity and conclusions of this work. Any use of trade, firm, or product names is for descriptive purposes only and does not imply endorsement by the U.S. Government.

**Data availability**

Data used in this manuscript are freely available. Raw weather data from the Sitka Airport are available through the University of Utah's MesoWest climate data portal for station PASI (https://mesowest.utah.edu/) with full records available through a Synoptic Data Service API (https://developers.synopticdata.com/mesonet/). The full Tongass National Forest landslide inventory (initiation points and landslide areas) is available from the U.S. Forest Service data portal (https://gis.data.alaska.gov). Processed data are available through GitHub (https://github.com/pattonai/sitka-lews) with a release on January 5, 2023 (doi:10.5281/zenodo.7508537).

**Author contributions**

A.I.P. and J.J.R. conceptualized the study and A.I.P. and L.V.L. co-led the study. A.I.P. and L.V.L. performed statistical analyses, produced figures, and co-wrote the text. J.J.R., A.J., O.K., and B.B.M. contributed to discussing results and reviewing the manuscript. L.V.L. and A.I.P. revised and edited the text. The authors have no competing interests.

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
