# Peer review of "Landslide initiation thresholds in data sparse regions: Application to landslide early warning criteria in Sitka, Alaska, USA"

_EGUsphere, 2023_

## Referee Comment (RC1)

The authors define rainfall thresholds for landslides for an area within 2 km from the road network and sensitive infrastructure in Sitka (Alaska). This is a highly localized dataset, with very few historical landslide events exists (5 days with events). They use hourly rainfall and compare different accumulation periods, different antecedent rainfall windows, different methods for the definition of the threshold (frequentist and Bayesian).

They consider both probabilistic predictions (probability of occurrence, logistic regression model) and intensity (number of landslides, Poisson regression model). They find the frequentist approach with 3h rainfall to be the best statistical model. They find antecedent rainfall not to sufficiently improve the performances of a triggering rainfall only threshold. Finally, they carry out robustness tests by leave-one-out and splitting the available record into calibration and testing sets.

I believe the manuscript is very well written and organized, and the development of a methodology which seems robust even with extremely few historical events is interesting and surely worth publishing in NHESS.

I only have few minor comments on the manuscript and one concern which is probably worth elaborating on/discussing in the manuscript.

I really appreciate all the work the authors put into verifying the robustness of the methods applied with such a limited number of historical landslide days. That said, I think some potentially critical aspects still could be further discussed. How representative are those events? I am thinking of two different things while posing this question which would both lead to overestimating triggering rainfall (by including only "extreme" events), the landslides used and the timing/triggering rainfall:

-   while the landslide record seems to be reconstructed from areal imagery (which would capture events regardless of whether there were damages associated or not), could there still be biases towards stronger/more damaging events?
-   How was the timing assigned? The authors mentioned eyewitnesses and precipitation record and based on Figure 2 it seems for all days except that of Harbor Mt. Slide the timing was well constrained (30min). But is precipitation used to constrain it to that time? Furthermore, the authors pick "the maximum cumulative precipitation in each sub-daily window". This could result in considering rainfall occurring a lot earlier than the actual landslide or, more importantly, after its occurrence. Previous studies already showed that it's typically not the strongest intensity that triggers landslides. E.g., Staley et al. (2013), looked at debris flow and showed that "*there were statistically significant differences between peak storm and triggering intensities*", confirming that it's not always the strongest rain to trigger them. If this is true, it could also apply to maximum cumulative rainfall.
    Figure 2 seem to suggest this shouldn't be a problem in general, but it's hard to really see the hourly intensities and still it could be the case e.g., for Harbor Mt. Slide where the maximum 3h cumulative is probably from 1h before the landslide time up to 1h after. This becomes potentially even more impactful if the rainfall record is used to narrow down the timing.

These aspects are important because while the authors really show the robustness of the method with respects of the available landslide events, having missed one event triggered by a small(er) amount of rainfall, could have a strong impact on the threshold (but also

possibly increasing the added value of considering antecedent rainfall). That would be the case if either some events have been missed/not reported or if the timing of any of the used events would be off by some hours. If the Harbor Mt. Slide happened 5-10 h earlier than the estimated time (the uncertainty is 12h) and only rainfall prior to that time was considered, how would it impact the results? This case study appears to be the best I have seen in terms of separation between rainfall on days with landslides or without, even for small domains, which could either be due to exceptional local properties (e.g., very homogeneous region) or indicative of the maybe non-representativeness of the landslide occurrence.

I really don't think any of these aspects invalidates the work presented or the methodology used, but it is probably worth discussing and adding some more information, especially about how the timing is determined and about whether rainfall after the estimated time is ever considered.

Finally, I have some very minor suggestions the authors could consider:
- Line 435: while 0.7 is a value commonly used to recognize a model that cannot be trusted, it could be interesting to report the Pareto-k values for the landslide days (since it's only 5 days)
- Comparison to "weighted coin toss": while this is presented as a baseline very simple approach and only used to report the BSS (and not a focus of the work presented), it would probably be more meaningful to compare to something more realistic (e.g., accounting for seasonality of the landslides, which all occurred in the August-November timeframe).
- While all components of the figures are explained in the captions, legend are usually helpful for the reader (e.g., landslide red lines in Figure 2, Figure 10).
- In Figure 10 the comparison among the plots is very difficult. While it clearly conveys the message that removing each landslide day does not have a strong impact, it might still be worth either replacing the 5 graphs (5 for probability, 5 for number) with just one where the all the estimated probabilities (and another for number of events) can be easily compared. Either removing the CIs or plotting only the edges with a different color (consistent with the probability) for each landslide removed.
- Figure 12 is a bit complicated to read. The authors could consider either splitting it into two different figures, because it looks like B would be a "zoom in" of A, or a calibration/test split, whereas they refer to different models. Furthermore, I would suggest removing the light blue area (instead just showing the edges, in an empty box around the timeframe in A and around the plot in B) and being more consistent in what is shown (e.g., in A and B the y axis show two different things). They might also remove the black vertical lines for events above the threshold(s). Finally, I am not sure what "*the gray field shows the 95% standard error*" refers to, but that probably will become more visible removing the light blue background.

Staley, D.M., Kean, J.W., Cannon, S.H. et al. Objective definition of rainfall intensity–duration thresholds for the initiation of post-fire debris flows in southern California. *Landslides* **10,** 547–562 (2013). https://doi.org/10.1007/s10346-012-0341-9.

---

## Author Comment (AC1)

**We thank both reviewers for their constructive reviews. We have prepared a draft of the manuscript with minor revisions, which we can provide to the editors if requested.**

Response to Reviewer 1

The authors define rainfall thresholds for landslides for an area within 2 km from the road network and sensitive infrastructure in Sitka (Alaska). This is a highly localized dataset, with very few historical landslide events exists (5 days with events). They use hourly rainfall and compare different accumulation periods, different antecedent rainfall windows, different methods for the definition of the threshold (frequentist and Bayesian).

They consider both probabilistic predictions (probability of occurrence, logistic regression model) and intensity (number of landslides, Poisson regression model). They find the frequentist approach with 3h rainfall to be the best statistical model. They find antecedent rainfall not to sufficiently improve the performances of a triggering rainfall only threshold. Finally, they carry out robustness tests by leave-one-out and splitting the available record into calibration and testing sets.

I believe the manuscript is very well written and organized, and the development of a methodology which seems robust even with extremely few historical events is interesting and surely worth publishing in NHESS.

I only have few minor comments on the manuscript and one concern which is probably worth elaborating on/discussing in the manuscript.

**We thank the reviewer for their encouraging review and constructive suggestions. As the reviewer notes, validating the robustness of a data-scarce model was a primary goal of this paper.**

I really appreciate all the work the authors put into verifying the robustness of the methods applied with such a limited number of historical landslide days. That said, I think some potentially critical aspects still could be further discussed. How representative are those events? I am thinking of two different things while posing this question which would both lead to overestimating triggering rainfall (by including only "extreme" events), the landslides used and the timing/triggering rainfall:
- While the landslide record seems to be reconstructed from areal imagery (which would capture events regardless of whether there were damages associated or not), could there still be biases towards stronger/more damaging events?

  **We of course cannot entirely rule out the potential for bias in the existing landslide inventory. However, based on our familiarity of the region and the Tongass inventory, we are confident that potentially impactful landslides are well-represented in the study area. It is possible that very minor slope failures or localized erosional features (a few square meters in area) are not documented, but these types of events with minimal potential for impact to safety or infrastructure are not the focus of our study. We do acknowledge the potential for some bias in the landslide inventory in discussion section 4.3.**

- How was the timing assigned? The authors mentioned eyewitnesses and precipitation record and based on Figure 2 it seems for all days except that of Harbor Mt. Slide the timing was well constrained (30min). But is precipitation used to constrain it to that time? Furthermore, the authors pick "the maximum cumulative precipitation in each sub-daily window". This could result in considering rainfall occurring a lot earlier than the actual landslide or, more importantly, after its occurrence. Previous

studies already showed that it's typically not the strongest intensity that triggers landslides. E.g., Staley et al. (2013), looked at debris flow and showed that "there were statistically significant differences between peak storm and triggering intensities", confirming that it's not always the strongest rain to trigger them. If this is true, it could also apply to maximum cumulative rainfall. Figure 2 seem to suggest this shouldn't be a problem in general, but it's hard to really see the hourly intensities and still it could be the case e.g., for Harbor Mt. Slide where the maximum 3h cumulative is probably from 1h before the landslide time up to 1h after. This becomes potentially even more impactful if the rainfall record is used to narrow down the timing.

**Thank you for this clarifying question. For events assigned a "precise" time in our manuscript (solid line in Figure 2), times were assigned based on eyewitness accounts or power outages (see details below). "Approximate" landslide times were estimated from the rainfall record, such that estimated landslide timing is ~1 hour after maximum hourly rainfall. These times were assigned manually for the sake of visualization in Figure 2, but are not relevant to the model set-up, which uses maximum 3-hour rainfall observed in each day. In a revised draft of the manuscript, we clarify in section 3.1 that precise timing is used for qualitative assessment, but that model set-up is less sensitive to precise timing information because we use maximum daily rainfall metrics.**

**The reviewer notes observations by Staley et al (2013) of post-fire debris flows triggered by non-peak rainfall. However, post-fire debris flows have unique characteristics and susceptibility, since they are triggered by infiltration-excess runoff. In our study, peak rainfall timing is well aligned with landslides for which precise timing information alone. Additionally, maximum daily rainfall reliably separates landslide and no-landslide events. This is consistent with other explorations of regional landslide data not presented in this manuscript, in which we found that landslide timing (when known) is typically within 1-3 hours following peak rainfall. Similarly, early studies by Roy Sidle found 2-6 hours.**

**In line section 2.2 of the revised manuscript, we clarify our assumption that triggering rainfall is well-represented by peak rainfall.**

**The timing of each of the landslides was assigned as follows. This information can be added to the supplemental document in a revised submission.**
- **The South Kramer landslide event on 8/18/2015 was assigned "precise" timing based on eyewitness accounts that stated the fatal landslide occurred at 9:30 am. We assume that other landslides during this storm occurred near that same time.**
- **The Halibut Pt landslide on 9/4/17 was assigned "approximate" timing at 12:00 pm based on a news report of a landslide that occurred "around noon" (https://www.kcaw.org/2017/09/04/landslide-closes-halibut-point-road-sitka/#:~:text=Officials%20in%20Sitka%20have%20closed,been%20evacuated%20as%20a%20precaution)**
- **The Medvejie slide on 9/20/19 was based on a news report of a power outage caused by the slide at "shortly before 1 pm." We therefore assigned the time to 12:50 pm. (https://www.kcaw.org/2019/09/20/slide-cuts-off-green-lake-road-hatchery-access/). The timing of the S. Kramer Landslide was assigned based on an eyewitness account which stated the time as 9:30 am.**
- **The Harbor Mountain landslide event on 10/26/2020 was labeled "approximate" because eyewitness accounts could only constrain the event to the night of occurrence. We assigned the time of "early morning" based on peak rainfall totals. Two landslides**

**occurred this night. The timestamp plotted on Figure 2 is estimated as occurring shortly after the timing of peak rainfall.**

○ **The Sand Dollar Drive landslide event included at least two periods of landsliding. Eyewitness accounts constrained timing to one "precise" event at 6:00 pm and one "approximate" event between 9:30 pm and 5:00 am (https://www.kcaw.org/2020/11/02/back-to-back-landslides-block-sitkas-sand-dollar-drive/).**

These aspects are important because while the authors really show the robustness of the method with respects of the available landslide events, having missed one event triggered by a small(er) amount of rainfall, could have a strong impact on the threshold (but also possibly increasing the added value of considering antecedent rainfall). That would be the case if either some events have been missed/not reported or if the timing of any of the used events would be off by some hours. If the Harbor Mt. Slide happened 5-10 h earlier than the estimated time (the uncertainty is 12h) and only rainfall prior to that time was considered, how would it impact the results? This case study appears to be the best I have seen in terms of separation between rainfall on days with landslides or without, even for small domains, which could either be due to exceptional local properties (e.g., very homogeneous region) or indicative of the maybe non-representativeness of the landslide occurrence. I really don't think any of these aspects invalidates the work presented or the methodology used, but it is probably worth discussing and adding some more information, especially about how the timing is determined and about whether rainfall after the estimated time is ever considered.

**This is an important question that we have addressed in several ways. First, we train the model on maximum daily rainfall totals, not rainfall preceding the landslide, so potential error in landslide timing does not impact model results. Additionally, evaluating model robustness in the case of missing landslide events is the primary goal of our leave-one-out analysis presented in Figure 10 (first column) and Supplemental Figure S1. We find that excluding the lowest landslide event from the model training results in surprisingly similar parameter estimates. As Bayesian analysis demonstrates, (Figure S1) excluding the low-rainfall landslides does not substantially change the median posterior parameter estimates, but the uncertainty for those values does increase. However, we recognize that sparse datasets may not always capture events in the tails of a distribution (for example, a landslide occurrence at low rainfall rates). This is part of what motivated us to select a conservative lower warning threshold at probability = 0.01, rather than the value which maximizes precision and recall (Fig. 11, first paragraph of section 3.4).**

**Based on the Reviewer's comment, we have now performed a counterfactual scenario analysis in which we *added* a hypothetical landslide at a lower rainfall value than landslides have been observed in the past (3-hourly rainfall = 18.0 mm), and re-fit the frequentist logistic regression (FL-3H-CF). We seek to test the sensitivity of our regression results to a potentially "missed" lower intensity landslide observation. We compared these results with FL-3H, the logistic regression model on which we based the thresholds. We find that parameter estimates change little (Figure AC1), although the uncertainties are reduced by virtue of having 6 landslide events instead of 5. From a practical perspective, the 3-hourly precipitation value associated with a probability of 0.01 (lower threshold) is 19.5 mm from this model, compared to 21.3 mm from FL-3H. 35.5 mm gives a probability of 0.7 (upper threshold), compared to the 34.0 mm used in the warning system. We surmise that in the unlikely case that a landslide that occurred at a lower rainfall value went unreported, it would not have impacted the parameter estimates in a way that is meaningful for the warning system. This robustness likely results from the relatively large number of days with lower rainfall values on which no landslides were reported. We can add details for this case in the supplemental file of a revised manuscript.**

[Figure]

**Figure AC1. "Missed" landslide counterfactual scenario.** (A) Results of FL-3H, including the five reported landslide events in Sitka (red points). (B) Counterfactual scenario with an additional landslide event (orange square) at 18 mm in three hours (FL-3H-CF). The orange line shows the results of FL-3H-CF, the red dashed line FL-3H. The confidence intervals refer to FL-3H-CF. (C) Comparison of parameter estimates for FL-3H-CF and FL-3H. Error bars show 95% confidence intervals based on standard errors.

Finally, I have some very minor suggestions the authors could consider:
- Line 435: while 0.7 is a value commonly used to recognize a model that cannot be trusted, it could be interesting to report the Pareto-k values for the landslide days (since it's only 5 days)

  ==We will report the Pareto-k values for the landslide days in section 3.4 of the updated version of the text.==

- Comparison to "weighted coin toss": while this is presented as a baseline very simple approach and only used to report the BSS (and not a focus of the work presented), it would probably be more meaningful to compare to something more realistic (e.g., accounting for seasonality of the landslides, which all occurred in the AugustNovember timeframe).

We consider our simple model a useful baseline for comparison, as it offers the simplest alternative to a precipitation-based model. This is similar to a common evaluation strategy in weather forecasting, in which forecasts are compared to climatology (e.g. Wilks, 2011; see more detailed response to Reviewer 2). This step also allowed us to evaluate if our model performed better than "random" guessing, but for rare events a comparison against 50:50 isn't particularly meaningful. However, instead of our historical frequency (5 / 6,606 = 0.0007), we could compare against landslide frequency on days in the rainy-*est* season (most ARs occur in August, September, October and November, but heavy rainfall can occur any time of year). In this case we obtain a daily frequency of 5 / 2215 rainy-season days in our record = 0.002. The BSS between FL-3H and wet season daily frequency is 0.54, indicating that our model still offers notable improvement.

- While all components of the figures are explained in the captions, legend are usually helpful for the reader (e.g., landslide red lines in Figure 2, Figure 10).

  Thank you for the suggestion, we added legend items to describe the red lines in Figure 2 and added a legend to Figure 10.

- In Figure 10 the comparison among the plots is very difficult. While it clearly conveys the message that removing each landslide day does not have a strong impact, it might still be worth either replacing the 5 graphs (5 for probability, 5 for number) with just one where the all the estimated probabilities (and another for number of events) can be easily compared. Either removing the CIs or plotting only the edges with a different color (consistent with the probability) for each landslide removed.

  Thank you for this suggestion. We experimented with this visualization and found that overlaying 5 model fits, particularly with the confidence intervals, becomes too busy to easily interpret. We prefer our layout which allows readers to compare how the model changes when specific landslide events are omitted.

- Figure 12 is a bit complicated to read. The authors could consider either splitting it into two different figures, because it looks like B would be a "zoom in" of A, or a calibration/test split, whereas they refer to different models. Furthermore, I would suggest removing the light blue area (instead just showing the edges, in an empty box around the timeframe in A and around the plot in B) and being more consistent in what is shown (e.g., in A and B the y axis show two different things). They might also remove the black vertical lines for events above the threshold(s). Finally, I am not sure what "the gray field shows the 95% standard error" refers to, but that probably will become more visible removing the light blue background.

  Thank you for this feedback. We have prepared a revised version of figure 12 in which the panels are more clearly differentiated to clarify that they show different values, the blue shading has been removed, and the gray shading is more visible.

Staley, D.M., Kean, J.W., Cannon, S.H. et al. Objective definition of rainfall intensity–duration thresholds for the initiation of post-fire debris flows in southern California. Landslides 10, 547–562 (2013). https://doi.org/10.1007/s10346-012-0341-9.

**References (for both author replies)**

Betancourt, M.: A Conceptual Introduction to Hamiltonian Monte Carlo, https://doi.org/10.48550/arXiv.1701.02434, 15 July 2018.

Crameri, F. (2018). Scientific colour maps. Zenodo. http://doi.org/10.5281/zenodo.1243862

Kuha, J., 2004, AIC and BIC: Comparisons of assumptions and performance: Sociological Methods and Research, v. 33, p. 188–229, doi:10.1177/0049124103262065.

Wilks, D. S.: Forecast Verification, in: Statistical Methods in the Atmospheric Sciences, vol. 100, Elsevier, 301–394, https://doi.org/10.1016/B978-0-12-385022-5.00008-7, 2011.

---

## Author Comment (AC2)

**We thank both reviewers for their constructive reviews. We have prepared a draft of the manuscript with minor revisions, which we can provide to the editors if requested.**

Response to Reviewer 2

**General comments**

This study deals with the relevant problem of establishing an early-warning system at a small regional scale with few observations. This problem is approached by testing different models with different inputs and tested for robustness. I think the study will be interesting to early-warning system developers as this system has been implemented in practice and therefore had to tackle many practical problems from how to establish thresholds to how to issue warning levels.

Generally, the manuscript is well written and organized. My main criticism is that the comparison/evaluation/validation is at times confusing and not as streamlined as other parts of the manuscript. Although I very much appreciate that much effort is put in the validation, I think section 2.3 (and 2.4 maybe) need some justification why so many different approaches are being taken. These should maybe also be presented more clearly later. You may have good reasons for choosing many different validation methods (leave-one-out for events, train/test split, different criterion metrics, etc.) but it's not clear to me from the text and I think it will be confusing to readers. For example, there is leave-one-out for landslide events and train/test split. Wouldn't it be simpler and similar added value to do leave-one-out with the recorded years (e.g. train with 2002-2018 and test with 2019)? Or why do you need the Brier skill score? Can't you use the same skill score as for the other models and compare these? Anyway, I don't think you go into details with the results of this part so it could be cut. To systematically compare the predictive power, you could also compute the area under the curve and assess if a model is better than random, as it is commonly done to assess predictive model skill.

I think if these issues could be solved, the manuscript will be more accessible to readers and that the authors are in a good position to solve this. Please find more specific comments below.

**We thank the reviewer for their positive assessments that our study will be interesting to early-warning system developers and that our manuscript is well written and organized.**

**We appreciate the constructive feedback that the motivation for some steps of our approach are not sufficiently clearly described.  Our model selection and evaluation strategy can be grouped into two parts. The first part concerns only the statistical models and their predictions. The second part treats the decision boundaries and their performance.  We argue that by keeping the statistical and decision making parts of the analysis separate, we are able to provide more transparency and information for decision makers (see, for example, https://hbiostat.org/blog/post/classification/index.html).  We elaborate on the motivation for each step below, and will update the text accordingly.**

*Statistical models*

1. **Model selection. First, we used information criteria (AIC, BIC, LOOIC) to select the most appropriate rainfall timescale from a range of possible options.  This step was necessary to decide which rainfall metric to use for the LEWS.  AIC and BIC both assess goodness of fit while penalizing complexity for the frequentist models, but have different**

drawbacks. Using both overcomes some of these limitations. LOOIC assesses out of sample predictive accuracy for the Bayesian models. The agreement between all three criteria that the 3-hourly timescale is most appropriate gave us additional confidence in our choice to use this timescale for the LEWS.

2. **Model evaluation.** After selecting the timescale, we evaluated the 3-hourly models (FL-3H, BL-3H, FP-3H, BP-3H) further. This consisted of two steps:

   a. We performed leave-one-out cross validation to assess how sensitive the parameter estimates were to individual landslide events. Since AIC, BIC, and LOOIC evaluate fit across the entire dataset (including the many non-landslide days), we wanted to specifically check the sensitivity to a potentially missed landslide event, and to assess how well the model might be able to predict a not yet observed landslide event. Because we estimate daily probability, we argue that leaving out a single day is a more appropriate check than leaving out an entire year, as the reviewer suggests.

   b. We used the BSS to compare the FL-3H model's skill to a simpler alternative model based on historical frequency. This is the first step of our workflow that evaluates skill. This approach is akin to a common strategy for evaluating weather forecasting models, in which the model's skill is compared to climatology: average weather conditions over long time periods (e.g. Wilks, 2011). In our case, climatology is replaced with historical landslide frequency. We note the important distinction between evaluating a model's predictive skill vs. evaluating its skill as a classifier. The Brier Score checks the predicted probability against the outcome (e.g. model predicts 0.8, and landslide occurs). This provides us with different information than the AUROC, which evaluates the model's skill as a classifier, and is thus a complementary metric. Classification requires choosing a decision boundary, which we will discuss below, but by providing and evaluating predicted probabilities, we offer more information for decision making than by only providing a classification. Therefore, we disagree that this section should be cut, but we will clarify the motivation for it in the updated manuscript, and present the results more clearly.

*Decision boundaries*

3. **Threshold selection.** Based on the statistical modeling results (FL-3H) and input from the Sitka community, we selected decision boundaries for implementation in the warning system by evaluating precision and recall. The reviewer suggests computing the AUROC to assess whether the model is better than random. The AUROC for FL-3H is 0.9993, and we include the ROC curve below (Figure AC2). This metric indicates that FL-3H far outperforms a classifier with no skill, and indeed is a near perfect classifier. However, in a strongly imbalanced dataset with near perfect separation between landslide days and non-landslide days like ours, we argue that the AUROC is a somewhat misleading metric, as its high value partially results from the large number of potential thresholds with a true positive rate of 1. While we could report this, it would be yet another step, which we do not feel adds much information. While the ROC curve can help to choose an optimal threshold, we argue that the Precision-Recall curve is more useful for this application in our case, because it better illustrates the tradeoffs between missed alarms and false

alarms (Figure 11). The optimal threshold found with the ROC curve (upper left corner) is identical to the threshold that maximizes recall.
4. Threshold evaluation. Finally, we evaluate the threshold performance by splitting the dataset into training and test sections. This is necessary to understand how well the warning system might perform in the future. In this case, we wanted to evaluate how many true alarms / false alarms / missed alarms / true no alarms would result from using the thresholds during the test period, rather than test the sensitivity of the statistical models to individual landslide days.

We have clarified these three separate goals for statistical evaluation in the abstract (lines 18-20). We have updated the methods introduction, as well as sections 2.3 and 2.4 to better explain and motivate each step. We have also added a more thorough treatment of the BSS results to the results in section 3.3.

[Figure]

**Figure AC2. ROC curve for FL-3H.** The dashed line represents a model with no skill for comparison.

**Specific comments**

L17: Please specify «136 statistical models». When only reading the abstract I cannot imagine what that means. How do models differ?

**We mention in the abstract that the 136 models have different timescales of precipitation variables. In our manuscript revision, we can clarify the wording of this statement. Further detail is provided to the readers in the methods section.**

L18: does that mean that your data is in daily resolution? If not, I would count the number of non-triggering rainfall events instead of the days.

**Our models are based on cumulative precipitation over a range of timescales and estimate daily landslide probability and intensity, therefore, we report the number of days. Our method avoids the need to define rainfall "events."**

L23-25: seems more like a conclusion to me. I would state this later in the abstract.

**In our revised manuscript we have moved this conclusion later in the abstract.**

L34-72: the intro nicely shows that there is a need for a LEWS in this region but that it's difficult to establish with current methods

**Thank you!**

L53-54: I think this sentence should end with a citation

**Several studies exist which can support this statement ("Accurately predicting rare events like landslides remains challenging because the complex and spatially heterogeneous processes that drive landslide initiation are difficult to characterize at sufficiently high resolution across broad regions") We add a few citations in our revised manuscript.**

L153: general question: what is the added value of the number of landslides if you don't know where it's going to happen?

**We estimate the number of landslides to provide information on whether widespread landsliding with multiple failures could be expected, or whether isolated landslides are more likely. This is valuable information for planning, even if the exact locations of the landslides aren't modeled.**

L217: So 1-day antecedent precip is the total daily rainfall at the day of the landslide?

**1-day antecedent precip is the total daily rainfall on the day *before* the landslide.**

L230: Please give a citation for these equations

**We have added a citation to the beginning of section 2.2 (McCullagh and Nelder, 1989, Generalized Linear Models) which describes both models.**

L235: Please integrate this sentence in another paragraph where you discuss this problem

**We have integrated this sentence into section 2.4.**

L247: more intuitive than what?

**Added that we mean more intuitive than frequentist confidence intervals.**

L271: What is a chain?

**Hamiltonian Monte Carlo, the sampler used to estimate the Bayesian posterior distributions, relies on Markov chains. The sampler progresses through the parameter space from point to point: a Markov chain is the record of these points (e.g. Betancourt, 2018). Essentially, what we mean when we say that we ran four chains for 2000 iterations is that four independent instances of the sampler progressed through 2000 points. We have specified that we mean Markov chains in the text.**

L276-279: how are these criterions defined?

**We have added citations to the text in section 2.3 which present these criteria in detail.**

L275-289: why was not the same validation performed on each model?

**After selecting the three-hourly time scale from among all considered models, we focus our efforts on evaluating these models for the sake of simplicity and timely production of scientific products.**

L332.335: I agree about the inadequacy of accuracy in imbalanced datasets but ROC shows exactly true alarms and false alarms in relative terms.

**We will remove the reference to ROC from this sentence.**

Fig. 3&4: please make sure these figures have the same layout (axis limits, labels, font size, order of plotting lines, etc.) (same for fig. 5&6). As these figures look very similar, you could consider e.g. only showing figs 3&5 here and move the others to the supplement.

**We find it informative to be able to compare the results of frequentist and Bayesian analysis, so we prefer to keep figures 3 & 6 in the main text. We have prepared modified versions of figures 3-6 to ensure that they have the same color scheme and font.**

Fig. 7/8: blue stands for better and red for worse. Better and worse compared to what?

**"Better" and "worse" refer to the AIC and BIC values used for model selection, where lower values (lower predicted error) are preferred (Kuha, 2004). The red and blue coloring is simply intended to help visualize the values in the tables for readability. Following your previous suggestion, we have added citations for AIC and BIC to the main text. We also adjusted the captions to indicate that model fit relates to estimated prediction error.**

Fig 11: Generally a very nice figure. Some comments:

**Thanks!**

- Maybe the color scheme for the landslide probability could be optimized, e.g. change in color where you set your threshold (at values mentioned in          L508)
    - **The color scheme is "batlow," a scientific color map that is optimized to be perceptually uniform and universally readable (Crameri, 2018). We prefer to mark**

**the threshold transitions using symbols instead.**

- Why is the lower threshold not slightly higher where recall is still 1?
  - **We chose to lower the threshold below the optimal threshold as a conservative approach to account for uncertainty in the probability estimates, as described in section 3.4 and discussed in 4.3.**

- When following this line from left to right, are you sure you can increase recall and precision at the same time? This is a          univariate model, right? I can't think of how this would happen. Since you have 5 events, should't the steps be in intervals of 0.2 for precision?
  - **Yes, it is possible to increase recall and precision at the same time in this univariate model. For example, at the step where the number of true alarms increases from 3 to 4, the number of false alarms stays at 7, and the number of missed alarms decreases from 2 to 1.  This leads to a simultaneous increase in recall from 0.6 to 0.8 and precision from 0.30 to 0.36.  With 5 events, the steps for recall (TA/(TA+MA)) are 0.2, but the steps in precision (TA/(TA+FA)) are influenced by the number of false alarms as well.**

- Caption: I would simply refer to the equations for definitions of recall and precision, but there you could mention the alternative names (e.g. precision=true positive rate)
  - **Thank you for the suggestion, we will modify the caption accordingly.**

L579: yes, compared to some shallow landslide thresholds this is rather low. Could it also be because some of the events are debris flows? The most predictive thresholds for runoff-triggered debris-flows can be at the 10-min timescale.

**Interesting question! Field observation suggests that the landslides in our study are shallow landslide failures which transitioned into debris flows. We have not observed runoff-generated debris flows in this study area. Similarly, we did not observe any signs of overland flow in the study area, but there are other small, shallow landslide scars in the region.**

L610: please specify "hydrologic monitoring". In this context, I assume soil moisture measurements.

**We can specify in section 4.1 that hydrologic monitoring includes soil moisture, groundwater level, and soil water potential.**

L619: I would say with "few landslide events" instead of "without". I don't think you investigated threshold determination without triggering events.

**We have modified this language in section 4.2 to refer to "few landslide events."**

L620-L625: you are of course right that negative events should be considered and in practice it may still be done only occasionally. However, this has been well-known for a while. The first ones I can think of are Staley et al. (2017, https://doi.org/10.1016/j.geomorph.2016.10.019) and Gariano et al. (2015, https://doi.org/10.1007/s11069-019-03830-x) and since then many others have adopted this procedure, some of them you cite earlier.

**This section intends to emphasize the potential for precipitation thresholds even when very few landslide events can be used to train the models, which is best reflected in the recent Peres and Cancelliere (2021) paper we cite here. The works by Staley et al. and Gariano et al. you note are good examples of landslide prediction; we added reference to Staley et al in section 1.3 (publication year for the link you included is 2017, *Geomorphology*). We cite Gariano et al. in section 4.1 (publication year for the link you included is 2020, *Natural Hazards*)**

L639-644: isn't this contradicting the earlier statement in L623-625 about the value of low precipitation totals? By using precision and recall you get rid of exactly these.

**As described in section 3.4, we selected thresholds using a heuristic approach which incorporated several sources of information, including Precision-Recall as well as a confusion matrix and qualitative assessment of risk tolerance.**

**References (for both author replies)**

Betancourt, M.: A Conceptual Introduction to Hamiltonian Monte Carlo, https://doi.org/10.48550/arXiv.1701.02434, 15 July 2018.

Crameri, F. (2018). Scientific colour maps. Zenodo. http://doi.org/10.5281/zenodo.1243862

Kuha, J., 2004, AIC and BIC: Comparisons of assumptions and performance: Sociological Methods and Research, v. 33, p. 188–229, doi:10.1177/0049124103262065.

Wilks, D. S.: Forecast Verification, in: Statistical Methods in the Atmospheric Sciences, vol. 100, Elsevier, 301–394, https://doi.org/10.1016/B978-0-12-385022-5.00008-7, 2011.